# The Macrophage–Fibroblast Dipole in the Context of Cardiac Repair and Fibrosis

**DOI:** 10.3390/biom14111403

**Published:** 2024-11-04

**Authors:** Stelios Psarras

**Affiliations:** Center of Basic Research, Biomedical Research Foundation of the Academy of Athens, Soranou Efesiou 4, 115 27 Athens, Greece; spsarras@bioacademy.gr

**Keywords:** cardiac fibrosis, tissue repair, macrophage, fibroblast, cellular interactions

## Abstract

Stromal and immune cells and their interactions have gained the attention of cardiology researchers and clinicians in recent years as their contribution in cardiac repair is increasingly recognized. The repair process in the heart is a particularly critical constellation of complex molecular and cellular events and interactions that characteristically fail to ensure adequate recovery following injury, insult, or exposure to stress conditions in this regeneration-hostile organ. The tremendous consequence of this pronounced inability to maintain homeostatic states is being translated in numerous ways promoting progress into heart failure, a deadly, irreversible condition requiring organ transplantation. Fibrosis is in fact a repair response eventually promoting cardiac dysfunction and cardiac fibroblasts are the major cellular players in this process, overproducing collagens and other extracellular matrix components when activated. On the other hand, macrophages may differentially affect fibroblasts and cardiac repair depending on their status and subsets. The opposite interaction is also probable. We discuss here the multifaceted aspects and crosstalk of this cell dipole and the opportunities it may offer for beneficial manipulation approaches that will hopefully lead to progress in heart disease interventions.

## 1. Introduction

The adult mammalian heart is a regeneration-hostile organ with a limited ability to repair itself following an injury. In a common and characteristic injury condition such as a myocardial infarction, ischemia-induced cardiac cell death triggers inflammation and molecular processes that are inefficient to orchestrate complete organ repair. In contrast, commonly emerging adverse remodeling features and scar formation may set a detrimental platform with a vicious circle of activated pathways leading to heart failure (HF). Similar processes may occur in other types of cardiac injury included in the multiple etiologies underlying HF. While cardiomyocytes constitute the critical problematic entity, being practically unable to re-expand and to substitute the damaged parenchyma, cardiac fibroblasts and macrophages vividly participate in the repair processes and their interactions may both direct the final cardiac output and comprise opportunities for beneficial manipulation in therapeutic interventions.

In recent years, our understanding of cardiac cell composition and stromal and immune cell contributions in health and disease has been dramatically improved owing to vast information generated by the exploitation of new analytical tools. Thus, single-cell or single-nucleus RNA sequencing (scRNASeq, snRNASeq) combined with spatial transcriptomics and multi-omics approaches revealed cell compositions and potential interactions primarily in human [1,2] and mouse [3] hearts, including the identification of cellular niches in specific cardiac structures [4]. These invaluable datasets reflect homeostatic states prevailing in the adult human heart, describe developmental changes [5], and expand towards specific cardiac repair and fibrosis conditions in cardiomyopathies leading to HF [6,7,8].

## 2. Cardiac Fibroblasts, Repair, and Fibrosis

Recent analysis confirmed previous notions suggesting that there are multiple fibroblast populations in the heart [3]. Additionally, various experimental tools and settings functionally discriminated against fibroblast subtypes, indicating that some of them exert opposite actions, including antifibrotic ones. While up to seven different fibroblast species appear to populate the adult human heart [1], further increasing in failing hearts [9], two main populations prevail in the murine heart originating from endocardium or epicardium [10]. Another classification identified periostin+ fibroblasts as being responsible for cardiomyocyte maturation and neuronal development in the peripartum cardiac tissue, while a less proliferative Tcf21+ population appeared to be preferentially dedicated to extracellular matrix (ECM) homeostasis [11]. Two major fibroblast populations appear to prevail also in the homeostatic human heart [9], however additional subsets were revealed in the steady state, as well in as the diseased heart, using RNASeq-based clustering [1,2,4,6,7,8,9,12]. Fibroblast subtype discrimination is important as it could allow specific targeting against detrimental species in future intervention strategies. The data from current efforts vary depending on the setting (see Table 1) and the whole approach is the subject of intense ongoing research.

In an injured and diseased heart, the fibroblast compartment expands and becomes activated. This has been successfully recapitulated in animal models, particularly in small rodents. Using cardiac injury models such as myocardial infarction (MI) and thoracic aortic constriction (TAC) it was shown that fibroblast populations expand in the murine injured heart [13] and become activated effectuating a pronounced deposition of collagens and other ECM components in a process markedly orchestrated by the signaling of transforming growth factor-beta (TGFβ) [14,15], a master fibrosis regulator. While fibroblast expansion combined with activation definitively sustains fibrosis, which in turn serves as a crucial component of cardiac rhythm irregularities [16] and HF development [17], it also represents a tissue repair process [18], which is necessary for ventricular wall integrity and cardiac function maintenance under conditions of massive cardiomyocyte loss. Indeed, when such pro-repair activities are compromised in cardiac fibroblasts, adverse cardiac remodeling, including fibrosis, is the outcome [19]. Notably, fibroblast proliferation capacity is inherent in mouse hearts right after birth, a developmental stage that supports efficient repair and regeneration following insults [20], but gradually declines to cease in the fibrosis-prone adult hearts [21]. In the desmin-null mouse, a genetic model of HF, cardiac fibroblasts show a defective phenotype with lower proliferation rates, increased apoptosis, and senescence that sustains repair inefficiency. When galectin-3, a trigger factor for this inability, was genetically deleted, several functional and structural parameters of the cardiomyopathy were improved partly due to the altered proliferative phenotype of cardiac fibroblasts [22]. Another regulatory circuit involves β2 adrenergic receptor (β2AR) stimulation that leads to extracellular signal-regulated kinase (ERK1/2)-mediated production of interleukin-6 (IL6) and to the proliferation of cardiac fibroblasts [23]. Under β2AR loss-of-function, the impairment in proliferation results in defective wound healing and increased MI infarcts. On the other hand, epigenetically driven loss of control leading to cardiac fibroblast hyper-proliferation impairs ECM balance and increases rupture after MI [24].

The fibroblasts share some characteristics with cardiac stem cells, particularly with mesenchymal stem (stromal) cells (MSCs) from which they may be indistinguishable in some respects including morphology [25], yet they may function differentially in cardiac fibrosis [26]. Other stromal stem cells like the PW1+ population possess the ability to differentiate into fibroblasts but also to induce cardiac fibroblast proliferation in a paracrine manner [27,28].

Genetic lineage tracing allowed for the discrimination of endocardium-derived fibroblasts from epicardium-derived fibroblasts in the TAC model. It was shown that specifically the fibroblasts of endocardial origin proliferate vividly, expanding from their main valvular location in steady state, and that they comprise in fact the subspecies responsible for pressure overload-associated cardiac fibrosis and dysfunction, in a process involving the activation of the Wntβ-catenin pathway [29]. Another mode of Wnt pathway involvement in TAC- or angiotensin II (AngII) infusion-mediated adverse remodeling relies on TEA domain transcription factor 1 (TEAD1) up-regulation in cardiac fibroblasts. TEAD1 interactions with the bromodomain containing protein BRD4 induces Wnt4 expression and β-catenin nuclear translocation, events which in turn promote collagen expression. Indeed, inhibition of TEAD1 interaction with Yes-associated protein (YAP) or its conditional deletion from cardiac fibroblasts reduced histological fibrosis and improved cardiac function in the TAC and AngII pressure overload models [30]. Similarly, YAP, which is being up-regulated in cardiac fibroblasts upon MI or neuroendocrine stimulation injury, binds at TEAD recognition sites to myocardin-related transcription factor A (MRTF-A) to induce fibroblast trans-differentiation into myofibroblasts, a process crucial for fibrosis development. When fibroblasts were deficient of YAP, the MI caused less fibrosis and dysfunction [31].

An additional pathway responsible for profibrotic fibroblast activation is the mitogen-activated protein kinase (MAPK)/p38. Accordingly, the ionophore salinomycin, identified as a strong antifibrotic agent in a high throughput screen setting, suppressed p38 and Rho signaling in cardiac fibroblasts, and both MI- and AngII-mediated cardiac fibrosis [32]. Another high throughput screen revealed an ERK/MAPK-dependent, Smad2/3-independent, pathway activation that involves eicosanoid degradation as being critical for profibrotic activation of cardiac fibroblasts [33]. Importantly, inhibition of such pathways in human fibroblasts isolated from end-stage HF patients allowed for the deactivation of their profibrotic character, opening strong translational opportunities in a previously thought irreversible process [33,34].

TGFβ, AngII, Wnt, and additional neurohormonal activation-triggered pathways, along with epigenetic modifications, converge to induce profibrotic fibroblast activation in the heart [16,35]. This includes, but is not restricted to, generation of alpha-smooth muscle actin (αSMA)+ myofibroblasts, a collagen overproducing cell type, from resident fibroblasts [36]. In AngII-induced cardiac fibrosis, for instance, among six emerging fibroblast populations, two were identified as being fibrogenic, with those expressing cartilage intermediate layer protein 1 (CILP1) being the predominant profibrotic species [37]. In contrast, other fibroblast subtypes such as fibroblast-specific protein 1 (FSP1)+ or collagen triple helix containing 1 (CTHRC1)+ fibroblasts were identified as distinct identities in human tissue or in preclinical models and are considered as exerting reparatory activities [38,39]. Proangiogenic activities are responsible for these repair promoting actions, at least for the FSP1+ subspecies. Triggers such as the platelet-derived growth factor (PDGF)-AB, which specifically restrain myofibroblast trans-differentiation without affecting the proliferation rates of cardiac fibroblasts, confer cardiac protection in MI accelerating early scar formation and reducing inflammation [40].

Intriguingly, cardiac fibroblasts possess endogenous regulatory pathways that restrain their profibrotic activation. A recent optimized snRNASeq analysis identified the transcription factor (TF) PBX/Knotted 1 homeobox-2 (PKNOX2) as a critical regulator that restrains both Smad2 and profibrotic signaling, likely by directly binding to genes such as *Postn* and *Col1a1* [12]. PKNOX2 expression is down-regulated in the fibroblasts in biopsies from dilated cardiomyopathy (DCM) patients while loss- and gain-of-function experiments in mice (TAC model) showed augmented or reduced cardiac fibrosis, respectively, suggesting that it may serve as a strong, endogenous anti-fibrotic player.

However, even activation of myofibroblasts should not be considered as de facto detrimental. For instance, myofibroblast-specific deletion of Smad3 signaling in mice led to increased cardiomyocyte apoptosis and proinflammatory macrophage infiltration in the TAC model associated with accelerated systolic dysfunction. The underlying mechanism involved an exaggerated loss of matrix integrity due to activation of metalloproteases (MMPs) such as MMP-8 [41]. Moreover, MI myofibroblasts were shown to exert phagocytosis of apoptotic cells, a process known as efferocytosis, canonically attributed to macrophages, and leading to the production of anti-inflammatory cytokines. This was mediated by the overexpression of a receptor called milk fat globule epidermal growth factor 8 (MFG-E8) by cardiac myofibroblasts and involved interaction with αvβ5 integrin [42]. Consequently, MFG-E8 administration to mice undergoing MI reduced cardiac fibrosis and improved function. Another population of fibroblasts that express major histocompatibility complex II (MHCII) was identified in biopsies from HF patients suffering from DCM [43]. These antigen presenting fibroblasts show up-regulated interferon-gamma (IFNγ) responses and JAK-STAT pathway activation, but they also seem to exert a regulatory action in DCM-associated HF, communicating via anti-inflammatory ligand–receptor interactions with CD4+ T cells.

Another beneficial aspect of cardiac fibroblast function is exerted in developmental stages or organisms where cardiac regeneration is favored. This is exemplified in the pre- or peripartum mouse heart and the zebrafish heart. Thus, prenatal murine fibroblasts secrete fibronectin (Fn) to mediate cardiomyocyte proliferation via integrin β1 signaling [44] and deposit nephronectin and Slit guidance ligand 2 (SLIT2) ECM proteins to induce cardiomyocyte cytokinesis [45]. Both abilities are lost in postnatal cardiac fibroblasts. On the other hand, epicardial-derived fibroblast-like cells in the zebrafish heart secrete neuregulin 1 (Nrg1) to achieve a similar outcome and counteract fibrosis [46]. Notably, in the adult zebrafish, heart transiently occurring fibrosis due to fibroblast inactivation is a step indispensable for cardiomyocyte proliferation and cardiac regeneration [47].

Apart from their trans-differentiation to profibrotic myofibroblasts, resident cardiac fibroblasts show plasticity and can be differentiated into additional forms that promote organ damage and dysfunction. Several days following MI induction in the mouse, fibroblast-derived matrifibrocytes emerge that express thrombospondin 5 and chondroadherin [48]. Such forms lack αSMA expression and are also present in infarcted human hearts. A detrimental osteogenic potential of cardiac fibroblasts has been also demonstrated in various forms of myocardial injury [49], including the pressure overloaded murine heart where it appears to depend on miR-129-5p expression that targets profibrotic asporin and pro-osteogenic SPY-Box transcription factor 9 (Sox9) [50].

Several pathways, converging or not, have been suggested to play a significant role in profibrotic fibroblast activation, including trans-differentiation into myofibroblasts in both ischemic and non-ischemic cardiomyopathy conditions. Some examples include the BET-dependent activation of the transcription factor mesenchyme homeobox 1 (MEOX1), required in TGFβ1-mediated activation [51], and the nuclear translocation of G-protein coupled receptor 5, contributing to AngII-mediated activation [52]. Members of the same family may have different effects. For instance, glycogen synthase kinase (GSK)-3α mediates the activation of the MEK-ERK-IL11 axis in cardiac fibroblasts in a SMAD3-independent fashion to promote pressure overload fibrosis [53], in contrast to the GSK-3β isoform that mitigates post-MI fibrosis and profibrotic fibroblast stimulation in a SMAD-dependent manner [54]. The transcription factor AE binding protein 1 (AEBP1) was identified as an important mediator in fibroblast-to-myofibroblast trans-differentiation, initially based on scRNAseq analysis of DCM biopsies and confirmed in in vitro settings [55,56]. AEBP1, together with proline and arginine-rich end leucine rich protein (PRELP) and collagen XXII A1 chain (COL22A1) were identified in a screen test as fibroblast-specific profibrotic gene products that mediate trans-differentiation into myofibroblasts [6]. The screen output included the reduction in αSMA cytoplasmic expression upon Cas9-mediated knock out of the examined genes and it was also based in a precedent snRNAseq analysis that determined the set of genes characterizing fibroblast activation in DCM and hypertrophic cardiomyopathy (HCM) HF biopsies.

In an indirect way of promoting cardiac damage including fibrosis, fibroblasts exert multifaceted proinflammatory activities in response to cardiac insults. An inflammatory feature of cardiac fibroblasts in MI is the secretion of IL6. Adenosine monophosphate (AMP) is primarily generated by non-cardiomyocytes in MI and accumulated by fibroblasts because it cannot be further cleaved to adenosine, a process canonically undertaken by neighboring T cells whose adenosine overproduction triggers IL6 expression by cardiac fibroblasts, triggering an inflammatory cascade [57]. Such proinflammatory activation of cardiac fibroblasts mediated by the Ras-ERK pathway and leading to increased IL6 and monocyte chemoattractant protein-1 (MCP1) secretion may also occur during aging, promoting fibrosis [58,59]. On the other hand, inflammasome activation in cardiac fibroblasts is evident in ischemia-reperfusion preclinical models and the resulting interleukin-1 beta (IL-1β) secretion likely contributes to subsequent fibrosis and dysfunction [60]. These and other proinflammatory aspects of fibroblasts will be further discussed in the section examining their direct interplay with macrophages within the context of cardiac repair and fibrosis.

## 3. Cardiac Macrophages, Injury, and Repair

In recent years, the significant contribution of macrophages in cardiac homeostasis has been robustly established, demonstrating among others their crucial roles in cardiac vessel development, conduction system maintenance, and homeostatic mitochondria withdrawal from cardiomyocytes [61,62]. Similarly to fibroblasts, macrophages are comprised of heterogeneous cardiac subpopulations. They populate the cardiac tissue early in development, eventually comprising a mixture of species derived from primitive yolk sac, fetal, and adult monocyte progenitors [63,64]. Their heterogeneity dramatically expands following cardiac injury and subsequent repair [9], as multiple subpopulations newly emerge originating from the periphery or from bone marrow monocytes. While monocyte-derived macrophages are largely proinflammatory and likely promote cardiac damage, resident macrophages seem to exert certain cardioprotective actions at various levels. Expression of C-C motif chemokine receptor 2 (CCR2) is considered to reveal the monocytic origin of a given macrophage subset. Notably, there are several exceptions to the above “rules”.

Several attempts based on combinations of transcriptional profiling, spatial identification, and conventional flow cytometry approaches determined macrophage subpopulations that are resident to the heart. According to recent research, in the mouse, the cardiac macrophage signature at steady state reflects the presence of three distinct major subsets [65,66] and conforms well with the macrophage species prevailing in other organs, including the lung, the liver, and even the brain [67]. While two of these subpopulations are CCR2-negative and can be discriminated based on the high levels of expression of either lymphatic vessel endothelial hyaluronan receptor 1 (LYVE1) and T cell immunoglobulin and mucin domain containing 4 (TIMD4) or MHCII, the third one expresses both MHCII and CCR2, suggesting that CCR2 expression is not an absolute marker of macrophage colonization. Notably, recent analyses using high resolution clustering identify up to nine different macrophage “states” in the steady state murine heart [68].

The number of distinct subsets in relevant reports varies depending on the analytical tool used. For instance, snRNASeq-based analysis revealed increased resident macrophage sets in human hearts when compared to the scRNASeq approach [9]. Nevertheless, the number of subpopulations further increases in the injured heart. Accordingly, two major subsets were identified in healthy human hearts and up to five subpopulations/clusters were identified in the injured heart, in addition to two monocyte subsets [9]. In the mouse, following an injury such as MI, several new subsets appear being derived from Ly6C^hi^ monocytes, a cell population that also infiltrates the infarcted heart early post MI. According to a recent analysis [65] among the major macrophage subsets infiltrating infarcted mouse hearts, two proinflammatory populations express IL1β and interferon type I (type I IFN)-stimulated genes, respectively, whereas another two express triggering receptor expressed on myeloid cells 2 (encoded by *Trem2*) along with either osteopontin (OPN, encoded by *Spp1*) or growth differentiation factor 15 (encoded by *Gdf15*), probably representing subsequent forms of monocyte-to-macrophage differentiation. Trem2^hi^ macrophages predominate in late stages post MI and exhibit anti-inflammatory polarization characteristics, while TREM2 itself, a secreted member of the immunoglobulin superfamily, improved cardiac repair when injected in mice with MI [66]. Moreover, a distinct CCR2+ subpopulation expressing the transcription factor basic helix-loop-helix family member E41 (encoded by *Bhlhe41*) emerges specifically during infarct building up and seems to be also protective, as suggested by the increased MI damage observed under Bhlhe41 deficiency [69]. Other subtypes, particularly those characterized by the activation of type I IFN pathways, exert detrimental actions. In mouse MI, interferon-inducible cell (IFNIC) macrophages respond via the cGAS-STING pathway to DNA released by necrotic cardiomyocytes and accelerate inflammation leading to ventricular wall rupture and increased lethality [70]. This type I IFN activation in MI is initiated in bone marrow monocytes and is counterbalanced by the expression of the transcriptional regulator nuclear factor (erythroid-derived 2)-like 2 (encoded by *Nrf2*) in resident, CCR2-negative, macrophages [71]. Inhibition of the cGAS-STING pathway and protection against myocardial wall rupture is associated with the switch of the macrophage status from inflammatory to reparative in this case [70]. Additional subsets have been identified in injured murine hearts depending on the protocol used [72].

Macrophages appear to play a crucial role in the efficient repair of the injured heart, particularly during its regeneration-friendly developmental stage, i.e., in the neonatal heart [73]. Part of this activity is exerted by factors secreted by macrophages in injured hearts from 1-day-old neonatal mice, while an altered macrophage secretome in injured hearts from mice that are 1 week older cannot support cardioprotective angiogenesis and cardiomyocyte proliferation. The cardioprotective action seems to be mediated by secreted C-C motif chemokine ligand (CCL)24, amphiregulin (AREG), and cardiotrophin-like cytokine factor 1 (CLCF1) directly affecting the cardiomyocytes [74]. The protective action seems to also involve distinct macrophage subpopulations expressing IL13 type II receptor (IL13IIR) that emerge upon MI and respond to IL4 and IL13 cytokines, the latter being produced by type II innate lymphoid cells. This ability is lost in the adult heart, but when IL13 is exogenously provided, the reparatory macrophages protect the heart via efficient efferocytosis [75]. The ability to exert efferocytosis appears to be a critical parameter affecting the cardioprotective arm of macrophages. In MI, legumain (encoded by *Lgmn* and specifically expressed in resident cardiac macrophages) is overexpressed and regulates the formation of LC3-II-dependent phagosome facilitating efferocytosis, while diminishing MI-mediated cardiac damage. Accordingly, *Lgmn* deficiency in the mouse augments inflammatory CCR2+ macrophage infiltration post MI that together with reduced efferocytosis of apoptotic cardiomyocytes results in increased infarct size and cardiac dysfunction [76]. On the other hand, cardioprotective efferocytosis seems to depend on vascular endothelial growth factor (VEGF) type C expression by cardiac macrophages, as VEGFC deficiency, or pharmacological inhibition impaired lymphangiogenesis and exacerbated MI damage, augmenting scar size and dysfunction during the reparatory phase [77].

In stress-induced cardiomyopathy, resident macrophages appear to exert strong cardioprotective actions. Part of this can be attributed to distinct states of macrophages (TIMD4^+^MHCII^lo^) that expand by proliferation during hypertensive stress in the mouse. A careful scRNASeq-based examination of resident cardiac macrophages identified up to nine subsets that differentially expanded and responded to hypertensive stress (AngII infusion) without altering their core transcriptional clusters [68]. Specific deletion of the resident macrophages using inducible genetic targeting resulted in loss of cardiomyocyte compensatory growth under chronic stress conditions and worsened cardiac fibrosis and dysfunction. This was attributed to insulin-like growth factor 1 (IGF1) production by resident macrophages [68]. This was also the case in a genetic model of DCM and HF where the adaptive remodeling was mediated by the resident macrophages that increased their IGF-1 expression following the sensing of the cardiomyocyte mechanical stress via transient receptors potential vanilloid (TRPV4) and focal adhesions complexes [78].

In the cumulative doxorubicin cardiotoxicity model, resident macrophages exert a reparatory role and following an initial retreat phase they expand in a class A1 scavenger receptor and c-Myc-dependent manner, restricting cardiomyocyte apoptosis and cardiac dysfunction [79]. In contrast, infiltrating macrophages initiate cardiac damage in acute doxorubicin cardiotoxicity, likely by producing catecholamines that mediate p53 activation and mitochondrial damage and apoptosis, as shown in H9c2 cardiac cells in vitro [80].

In contrast to their resident counterparts, numerous detrimental properties have been attributed to macrophages that infiltrate the heart following a challenge, stress, insult, or injury. Thus, in cardiac transplantation, CCR2+ macrophages are responsible for allograft rejection via MyD88-mediated allograft-specific T cell activation, while macrophages not expressing CCR2 play a protective role as their depletion increased rejection rates [81]. In the genetic model of arrhythmogenic cardiomyopathy caused by a pathologic variant of desmoglein 2 (Dsg2^mut/mut^), the infiltrating CCR2+ macrophages confer NFκΒ-dependent dysfunctional changes to the cardiomyocytes [82]. Differential actions are exerted by infiltrating macrophages in the isoproterenol stress injury model. A single administration of high-dose isoproterenol causes β1-adrenergic receptor stimulation, inflammation, reversible cardiac dysfunction, and cardiac fibrosis in small animal models and recapitulates stress-induced cardiomyopathy conditions, including Takotsubo cardiomyopathy [83,84]. In this setting, depletion of macrophages or inhibition of macrophage infiltration and activation improved both cardiac fibrosis and dysfunction [84]. On the other hand, exposure to isoproterenol is reminiscent of cardioprotective preconditioning as a second administration of the drug fails to cause cardiomyopathy and this is again macrophage-dependent since macrophage depletion abolished protection [85]. Diverse effects on cardiac repair have been attributed to timely prevailing macrophage subspecies in the TAC model. In this model, resident macrophages expand initially by local proliferation and protect against damage by promoting angiogenesis, whereas the subsequently dominating CCR2+ monocyte-derived macrophages generate an antiangiogenic milieu compromising cardiac function [86]. Pharmacological and antibody-based inhibition of CCR2 reduced fibrosis, hypertrophy and dysfunction, and regulated additional cellular players mediating cardiomyopathy in this model, such as the expanding T cell populations [87]. Another study using Cite-Seq, scRNASeq, and CytOF technologies to discriminate resident from infiltrating macrophages found that it is the resident macrophage population that readily expands one week following TAC injury and that timely regulated blockade by colony stimulating factor 1 receptor 1 (CSFR1), which preferentially abolishes resident macrophages early after TAC, leads to extended fibrosis 6 weeks later [88]. Part of the protective role of resident macrophages in this setting was attributed to their ability to promote angiogenesis [88].

The abundance of CCR2+ macrophages has been also associated with adverse outcomes in human HF. A fraction of HF patients undergoing left ventricular assist device (LVAD) implantation while waiting for transplantation, positively responds to the burden relief of cardiac work and shows fortunate reverse remodeling changes in the myocardial tissue along with improved clinical scores [89]. In such LVAD responders, the levels of CCR2+ macrophages in cardiac tissue are lower than that observed in non-responders [90]. On the other hand, proliferating clusters of CD163+ resident macrophages are diminished in biopsies from HCM and/or DCM HF sufferers requiring transplantation [6,9].

Notably, proinflammatory macrophage activation may also confer cardioprotection under specific conditions. In fact, an adaptor known as interleukin-1-receptor-associated kinase-4 (IRAK-4) and acting downstream of proinflammatory MyD88 inhibits migration of inflammatory monocytes in experimental viral myocarditis. Under IRAK4-deficiency, the increased numbers of inflammatory monocytes infiltrating the heart following coxsackievirus B3 infection produce IFN-α and IFN-γ in quantities adequate to eliminate the virus and protect against damage [91]. Other extracardiac sources of macrophages such as the Gata6+ pericardial population may positively affect cardiac repair by inhibiting fibrosis as revealed by discontinuing their cardiac infiltration in MI models [92].

A pro-repair function of macrophages has been established under conditions where cardiac repair is exceptionally efficient. In the neonatal mouse heart, where for a limited time postpartum, the cardiac tissue can efficiently cope with injuries such as MI and local low temperature exposure or even amputation [20], the successful repair depends on the presence of macrophages that secrete angiogenic factors to sustain cardiomyocyte proliferation [73]. A central feature of this ability is the expansion of CCR2- (resident) macrophages in response to injury and the absence of CCR2+ recruitment in the neonatal hearts [63].

In contrast to the regeneration-friendly neonatal heart, the adult heart responds to injury by recruiting detrimental CCR2+ subsets [63]. On the other hand, up to 60% of the resident macrophages can be lost 2 days after an MI injury in an adult heart, and even 4 weeks later, the resident population falls short of its original levels [93]. A beneficial role of resident macrophages in this case has been demonstrated in settings allowing the depletion of this population, which in turn worsened the MI systolic dysfunction. Spatial specificity was also evident 2–4 days post MI, as the resident macrophages were particularly expanded by proliferation in the peri-infarct area of the murine hearts [93]. A time-dependent component is also a strong contributor in post-MI macrophage functions. Indeed, 1 day after injury, the macrophages, enriched by monocyte-derived infiltrating subsets, accommodate proinflammatory and ECM-degrading features. After 48 h, the macrophage compartment undergoes metabolic reprogramming promoting oxidative phosphorylation and exhibits proliferative and phagocytosis activities, while after 1 week, it is characterized by a pro-repair signature [94]. Manipulation of such early changes of the macrophage compartment can affect the cardiac outcome at later time points. Indeed, preferential depletion of resident macrophages before the application of pressure overload in the mouse (TAC model) led to increased fibrosis and reduced angiogenesis, accompanied by cardiac dysfunction 6 weeks later. Importantly, this worsening occurred despite the complete replenishment of the resident population at this time point [88]. In parallel to the resident macrophage kinetics, additional events, including efferocytosis, transform CCR2+ infiltrating macrophages into reparatory species in a time-dependent manner. In MI, for instance, the recruited CCR2^+^Ly6C^high^ inflammatory macrophages adopt a reparatory phenotype 3 days post injury [95]. In MI’s reparatory phase, recruited Ly6C^lo^ monocytes are also becoming anti-inflammatory CCR2^−^Ly6C^lo^ macrophages [96].

Thus, the macrophage compartment contains different subsets that may exert variable actions promoting either cardiac repair and regeneration or cardiac fibrosis and dysfunction. Significant time- and condition-dependent alterations in the composition of the macrophage compartment may critically affect the cardiac outcome. The precise identification and characterization of the macrophage subsets at different time points post injury and in human cardiac disease is thus of particular importance as it is expected to provide specific targets for future therapeutic interventions [97]. Its detailed description is, however, beyond the scope of this review and the reader should refer to several comprehensive reviews summarizing the current efforts [98,99,100].

The activation status of macrophages is also important. A classification of the macrophage subtypes, mostly popular in the past, included the so-called M1/M2 paradigm. According to this, macrophages may either adopt the “classical/conventional” proinflammatory (M1), or an “alternative”, anti-inflammatory, and presumably pro-repair (M2) phenotype characterized by differential expression profiles and surface molecule abundance. Although the existence of such polarized populations has been strongly debated, particularly in MI [70], and a spectrum of highly plastic activation states has been proposed to better reflect the in vivo conditions instead [101], the M1/M2 model is still useful to understand the complexities of macrophage activation status in ex vivo settings and can be applied to understand disease progress and facilitate in vivo manipulations, with some modifications [102]. An interesting example of the M1/M2 importance is that MI in regenerative hearts (from day 1 neonates) triggers a faster infiltration of macrophages and monocytes compared to that observed in anti-regenerative hearts (from day 8 neonates). In the regenerative hearts, the M2-like macrophages were found elevated 1 day post injury (accounting from 1.6% to 10.7%of the non-myocytes). In addition, the macrophages infiltrating the regenerative hearts expressed the secreted cardiotrophin-like cytokine factor 1 (CLCF1) that promoted neonatal cardiomyocyte proliferation [74]. Moreover, in the regeneration-prone heart of the zebrafish, delayed macrophage recruitment following clodronate liposome treatment impairs both neutrophil resolution and regeneration. Two populations of resident macrophages were identified with scRNASeq (hbaa+ and timp4.3+) that are necessary to support regeneration, and their absence cannot be compensated by circulating macrophages [103]. Moreover, in the even more rapid regeneration procedure occurring in zebrafish larvae, the efficient 48-h-long repair depends on macrophages that populate the epicardial-myocardial niche, expanding the epicardium that activates a vegfaa-notch axis to induce cardiomyocyte proliferation [104]. In neonatal mice deficient of IL4/IL13, two cytokines crucial for the anti-inflammatory potential and M2 activation of macrophages, cardiac regeneration is compromised; while this defect could be recapitulated by myeloid-specific deletion of *IL4ra*, that encodes the common IL4 and IL13 receptor mediating activation of the transcription factor STAT6, the major downstream mediator of this regenerative pathway [105]. Finally, in the regeneration-friendly zebrafish heart, proinflammatory (tnfa+) and reparative (tnfa-negative) macrophages are sequentially motivated to orchestrate efficient cardiomyocyte proliferation and cardiac repair [106].

Metabolic remodeling of macrophages, including the cardiac species, is an important feature of their pronounced plasticity and affects tissue repair after injury. While proinflammatory macrophages preferentially use glycolysis for ATP production and energy demands to cover their effector functions, reparative macrophages switch to mitochondrial oxidative phosphorylation, with their efferocytosis activities further supporting mitochondrial respiration [107]. In MI, macrophages show increased expression of glycolytic enzymes early after injury and increased expression of genes promoting mitochondrial oxidative phosphorylation at later time points [94]. These changes timely coincide with the early proinflammatory phenotype in contrast to the reparatory activity observed later. Mitochondrial homeostasis is important in these processes. Indeed, macrophages with deletion of *Ndufs4*, encoding a member of mitochondrial respiratory complex I, assume a proinflammatory phenotype with increased glycolytic activity and reduced mitochondrial function, while *Ndufs4* knock-out mice presented with suppressed efferocytosis and a delayed transition of proinflammatory to reparatory macrophages after MI, which was associated with increased scarring, thinner LV walls, and compromised repair after MI [108]. Mechanistically, the macrophages with a more pronounced proinflammatory phenotype due to the *Ndufs4* deficiency were inferior in their ability to paracrinely induce aSMA and collagen I expression in cardiac fibroblasts compared to wt macrophages during in vitro experimentation. Accordingly, lower myofibroblast formation and fibroblast proliferation rates were observed in *Ndufs4*-deficienct mice following MI, likely resulting in lower repair activities of fibroblasts, which led to the formation of a thinner myocardial wall [108].

Metabolic remodeling and epigenetic mechanisms converge to regulate the switch of cardiac macrophages to their reparative phenotype by several mechanisms. Nucleophosmin (NPM1) is up-regulated in MI and its oligomeric form recruits the demethylase KDM5b in the promoter of TMC complex subunit 1 (TSC1), reducing H3K4me3 demethylation and TSC1 expression. Because TSC1 serves as an mTOR inhibitor, the unleashed mTOR activity augments mTOR-related glycolysis in macrophages, thus inhibiting the switch to the reparatory phenotype [109]. Accordingly, the inhibition of NMP1 enhanced cardiac repair post MI by improving angiogenesis. On the other hand, adenosine kinase (ADK) expression increases in macrophages 7 days post MI. This enzyme regulates intracellular adenosine levels, allowing in turn H3K4me3 transmethylation in the interferon regulatory factor 4 (Irf4) promoter. Myeloid-specific deletion of *Adk* in the mouse resulted in decreased expression both of IRF4 and of molecules involved in the process of macrophage switch to the reparatory phenotype, leading to impaired fibroblast activation and to severe cardiac dysfunction in MI [110].

A member of the NOD-like receptors, NLRC5, which is up-regulated in cardiac macrophages in the TAC model, appears to be protective in pressure overload remodeling and HF. Indeed, *Nlrc5*-deficient mice showed worsened TAC-induced hypertrophy and fibrosis, and bone marrow-derived macrophages (BMDMs) deficient in *Nlrc5* were more prone to induce cardiomyocyte hypertrophy and myofibroblast trans-differentiation of cardiac fibroblasts [111]. Mechanistically, NLRC5 interacts with heat shock protein 8 in macrophages, controlling expression and secretion of IL6, a known mediator of cardiac hypertrophy and fibrosis.

In summary, both macrophages and fibroblasts are heterogenous and highly plastic cell populations whose functions strongly affect the cardiac repair process. Furthermore, depending on the cardiac injury and disease condition, and/or the time point considered, their actions support regeneration or promote fibrosis and dysfunction. Most of their main features described above were revealed from studies in small animal models of cardiac injury, particularly using the laboratory mouse, and are summarized in a comparative manner in Figure 1.

Importantly, the macrophage status may directly regulate fibroblast features affecting cardiac repair and fibrosis. Several lines of evidence further support the reciprocal interactions of these two cell types justifying their concerted action following cardiac injury and these will be deployed in more detail in the next sections. Moreover, a summary of this information can be found in Table 2.

## 4. The Macrophage–Fibroblast Dipole

Macrophages and fibroblasts can be found in close vicinity to each other in the cardiac tissue and likely interact. Several works using preclinical models have suggested mediators and ways of crosstalk within the fibroblast–macrophage dipole that affect cardiac repair and fibrosis. However, the majority of these are based on indirect studies, whereas settings examining direct interactions are difficult to apply, partly due to the heterogeneity and plasticity of the involved cell types.

The spatial vicinity of macrophages and fibroblasts has been demonstrated in diseased cardiac tissue [7]. This is crucial because vicinity is a prerequisite allowing the interaction between two cell types and is particularly important in the context of tissue repair and fibrosis [155]. In a recent example, the macrophage-to-fibroblast spatial vicinity in the damaged myocardial tissue of the desmin-null mouse (des^−/−^) seems to effectuate the profibrotic action of galectin-3, a beta-galactoside binding lectin that is being released by the cardiac macrophages [22]. Both secreted factors such as galectin-3 and direct cell–cell contacts would establish an intercommunicative network in the heart [156], in addition to mechano-transduction pathways involving ECM that could accommodate more remote attractions [157]. Differences in the expression patterns of chemokines, cytokines, growth factors, and ECM components observed among states of each cell type either favor efficient regeneration or promote impairment of repair programs ending up in fibrosis in the cardiac tissue. For instance, transcriptional alterations between day 1 postnatal and day 8 postnatal murine hearts undergoing MI injury refer to, and may reflect, altered interactions between cardiac macrophages and fibroblasts decisively affecting the outcome [158]. Nevertheless, the two cell types are characterized by distinct RNA or protein expression profiles [159] and their potential interactions in the process of cardiac repair are readily reflected in collective bioinformatic analyses based on scRNASeq data [160].

The importance of the macrophage–fibroblast dipole was evident in a recent snRNASeq analysis of end-stage HF patients beneficially responding to LVAD implantation. Compared with non-responders, patients with signs of reverse remodeling and cardiac recovery showed down-regulation of the transcription factor RUNX1 in both fibroblasts and macrophages [161]. In contrast, the transcriptional profile of cardiomyocytes remained unchanged, suggesting that alterations in macrophages and fibroblasts may specifically orchestrate cardiac recovery. Recapitulation of RUNX1 containment in pressure overload-associated HF in mice (TAC model) using bromodomain containing 4 (BRD4) inhibition led to cardiac recovery as well. This was accompanied by an increased ratio of resident (presumably reparatory) over inflammatory macrophages, paralleled by a reduction in activated periostin+ fibroblasts in favor of homeostatic ApoE+ fibroblasts in the mouse heart [161]. These data suggest that the RUNX1 pathway may set a common platform of the macrophage–fibroblast dipole contribution in cardiac repair.

## 5. Macrophages Affecting Cardiac Fibroblasts

Several preclinical settings were used to explore the direct effect of macrophages on fibroblasts occasionally supported by sparse clinical correlations (Figure 2). Among these, in a mouse model of robust cardiac fibrosis (AngII infusion), depletion of monocytes by clodronate liposomes ended up in reduced fibrosis, while cocultures of CD11b+ macrophages with murine fibroblasts induced αSMA expression in the latter, indicating a myofibroblast-like activation [112]. Moreover, earlier reports identified CD11b+ inflammatory cells (presumably macrophages) as resources of the major profibrotic agent TGFβ in human biopsies from HF with preserved ejection fraction (HFpEF) patients and attributed this event to the increased collagen I production and ability of biopsy-derived human cardiac fibroblasts to trans-differentiate into myofibroblasts. On the other hand, an observed MMP1 reduction was associated with further establishment of the fibrotic network [113] (Figure 2). Recent reports ascertain that in ischemic HF, TGFβ1 signaling is specifically activated in monocytes triggered by coagulation cascades [114] (Figure 2). This involves interaction of the tissue factor (TF) and the protease activator 2 (PAR2) at the cell surface and intracellular MAPK/ERK activation in CCR2+ myeloid cells. Pharmacological inhibition of both points results in reduced SMAD2 activation in cardiac fibroblasts, likely via diminished TGFβ released by the myeloid cells [114]. Moreover, profibrotic TGFβ along with IL-10 were found to be secreted by cardiac macrophages isolated from pressure-overload-stressed mice (TAC model), at levels like those mediating cardiac fibroblast to myofibroblast trans-differentiation when recombinant TGFβ and IL-10 were exogenously added [88]. These data indirectly suggest that cardiac macrophages mediate fibroblast profibrotic activation in the TAC model.

Other mediators have been also implicated in the macrophage-to-fibroblast signaling in cardiac repair (Figure 2). Early post-MI inhibition of the cGAS/STING axis in mice leads to a macrophage switch from the inflammatory to the reparatory phenotype (Figure 2), which is associated with increased generation of myofibroblasts, leading to denser collagen deposition in the ischemic damage that protects from rupture [70]. Elevated myofibroblast abundance can be also sustained by increased fibronectin (Fn) secretion by the macrophages themselves [115]. Recent analysis identified a population of monocyte-derived macrophages expressing osteopontin, Fn1, and arginase 1 (Arg1) that expands post MI and likely promotes fibroblast activation and fibrosis in a C-X-C motif (CXC)L4-dependent manner [116]. Indeed, a less profibrotic profile of this macrophage population, as judged by expression profiling, coinciding with a reduction in the expression of core matrisome components by cardiac fibroblasts, was evidenced in MI-injured CXCL4 knock-out mice. This work indirectly, yet strongly, suggests a contribution of the osteopontin-expressing subpopulation to fibroblast-mediated fibrosis in MI (Figure 2). Intriguingly, a similar population appears to exist not only in human HF but also to promote renal fibrosis in relevant experimental models, indicating a universal mechanism [116]. Moreover, CXCL4 signaling appears to mediate proinflammatory and profibrotic macrophage stimulation in the murine SAUNA model that shares some similarities with human HFpEF (Figure 2). In that case, myeloid-specific CXCR4 deficiency reduced cardiac fibrosis and diastolic dysfunction by lowering CXCL3 chemokine secretion. This protective effect was recapitulated in experimental settings where the conditioned medium from CXCR4-deficient BMDM was applied to neonatal cardiac fibroblast and lowered their ECM production and potential to trans-differentiate to myofibroblasts [117].

Moreover, in stress-induced or aging-developed interstitial fibrosis that has been associated with diastolic dysfunction, monocyte-derived macrophages expand and produce IL10 that partially mediates fibroblast activation, as revealed by reduced fibrosis observed under macrophage-specific *Il10* gene silencing [118]. Notably, while exogenous IL10 alone did not activate profibrotic expression by cardiac fibroblasts, it did so, when macrophage secretome was included in the experimental setting, while osteopontin that was co-produced by cardiac macrophages played a co-stimulator’s role [118]. Thus, IL10 and osteopontin of macrophage origin collaboratively promote cardiac fibroblast activation leading to fibrosis in this model (Figure 2). Fibrosis and cardiac dysfunction were indirectly mediated by osteopontin of macrophage origin in a genetic model of HF, the des^−/−^ mouse. In this case, we identified galectin-3 (Gal3, encoded by *Lgals3*) as being up-regulated by osteopontin (OPN) in cardiac macrophages [119] (Figure 2). Macrophage-secreted galectin-3 was essential in the activation of profibrotic pathways in the murine heart and in cultured cardiac fibroblasts [22]. In addition, galectin-3 up-regulation in macrophages was evident in the lungs of this animal model, where it appeared to partially mediate an emphysema-like alveolar dilation and lung dysfunction. Both the cardiac dysfunction and the pulmonary co-morbidity were reduced under loss-of-function of galectin-3 in the des^−/−^ mouse [22]. Moreover, an inhibitor of galectin-3 conferred cardioprotection in a rat MI model by altering macrophage polarization, as well as Arg1 and IL10 expression, which, in the long term, led to a reduced fibrosis and improved cardiac function [162].

OPN-producing CCR2+ macrophages seem also to be involved in atrial fibrillation (Figure 2). Local fibrosis is essential for the development of arrhythmias in this cardiac chamber [16]. In a mouse model of atrial fibrillation (the HOMER mouse), scRNASeq revealed the expansion of SPP1+CCR2+ monocyte-derived macrophages as a crucial event in rhythm abnormalities [120]. Indeed, Spp1^−/−^ or Ccr2^−/−^ mice showed attenuated atrial fibrillation while macrophage OPN was identified as a mediator of TGFβ1-dependent inflammatory and profibrotic activation of atrial fibroblasts, both in the mouse model and in atrial fibrillation biopsies [120].

The action of proinflammatory IL-1β is central to cardiac repair and HF and its direct inhibition was the first large-scale anti-inflammatory therapy with clearly cardioprotective results, demonstrated in the CANTOS phase III trial [163]. IL-1β appears to be also an important mediator of macrophage- and macrophage-to-fibroblast signaling in cardiac repair (Figure 2). In a model of high-fat diet (HFD)-induced HFpEF, for instance, CD86+ or CCR2+ inflammatory macrophages increase whereas anti-inflammatory subsets decrease in the cardiac tissue. The inflammatory macrophages are responsible for IL1β overproduction that confers detrimental mitochondrial oxidation to cardiomyocytes, while depletion of these macrophages leads to a reduction in IL1β levels and mitigation of HFpEF [121]. At the molecular level, BRD4 binds to enhancers proximal to *Il1b* co-regulating with p65/RELA and its expression in C-X3-C motif receptor 1 (Cx3Cr1)+ myeloid cells under conditions of cardiac stress [122]. IL1β induced in turn a myofibroblast phenotype in fibroblast-like cells (generated from human pluripotent stem cells in that case) further augmenting TGFβs’ action. The profibrotic IL1β action was mediated again by p65/RELA that was bound at the enhancer of MEOX1, leading to its activation (Figure 2). As mentioned above, MEOX1 is a transcription factor important for profibrotic fibroblast activation [51]. This profibrotic effect was recapitulated after exposure of fibroblasts to the secretome of bacterial lipopolysaccharide (LPS)-activated BMDM and diminished by an IL1β neutralizing antibody [122]. Accordingly, BRD4 deletion in Cx3Cr1+ myeloid cells reduced IL1β levels and fibrosis, protecting against HF in the TAC model of cardiac stress [122].

Another cytokine important for cardiac pathophysiology, IL6, was shown to be elevated following the co-culturing of cardiac fibroblasts and macrophages. IL6 up-regulation mediated TGFβ1 overexpression and Smad3 activation by the fibroblasts (Figure 2), apparently promoting cardiac fibrosis in the AngII infusion model [123]. IL6 plays a detrimental role in MI with fibroblasts serving as a major IL6 source in humans and in mouse models, while T cells are also involved in the regulation of IL6 expression [57].

MiRNAs or other non-coding RNAs commonly contained in the exosomal compartment of macrophage secretomes are also important contributors in cardiac repair and regeneration [164] and mediate instructions from macrophages to fibroblasts (Figure 2). Indeed, when miR-21, the most highly expressed miR species in cardiac macrophages favoring both M1 macrophage activation and fibroblast-to-myofibroblast trans-differentiation, has been specifically knocked-out in macrophages, both events fainted rescuing TAC-injured mice from fibrosis progression and HF [124]. Within this context, in vitro paracrine M1-activation of BMDMs was shown to induce cardiac fibroblast trans-differentiation in a miR21-dependent manner [124]. A different way of action is exerted by miR-155 found in macrophage exosomes in MI (Figure 2) [125]. When cardiac fibroblasts uptake the exosomes, their proliferation rates are suppressed via differential regulation of son of sevenless-1 (*Sos1*) expression. This, however, is a defective repair event in the proliferative phase in MI, resulting in increased rates of cardiac rupture in early post-MI time points [125]. In parallel, inhibition of the suppressor of cytokine signaling 1 (encoded by *Socs1*) is another consequence of the miR-155 containing exosome uptake, resulting in augmented post-MI inflammation (and cardiac rupture) because of induced expression of proinflammatory cytokines (IL1β, TNFα, IL6, MCP1) which are canonically inhibited by SOCS1 [125].

However, other mediators secreted by macrophage subsets may act differently in cardiac fibrosis (Figure 2). Thus, in early time points following TAC injury, accumulating LY6C^hi^ macrophages in hypoxic areas release oncostatin M (OSM) that inhibits myofibroblast trans-differentiation by phosphorylating the SMAD linker region and protects from fibrosis [126]. Moreover, the cardioprotective action of the recently identified Bhlhe4+ subpopulation of macrophages in MI has been partially attributed to their ability to detain αSMA expression by fibroblasts, as shown in co-culture settings [69]. It has been proposed that progranulin, overexpressed by this macrophage subset, interacts with TNFR1 antagonizing anti-repair actions of tumor necrosis factor-alpha (TNFα) in the fibroblasts (Figure 2). In a similar context, a subset of macrophages activated by a1 adrenergic signaling (adra1) in an injury model of zebrafish larvae adopt a reparative (tnfa-negative) phenotype that differentiate zebrafish cardiac fibroblasts into a collagen 12-expressing phenotype, which in turn efficiently drives cardiac regeneration. This regeneration-friendly crosstalk is mediated in a paracrine manner by midkine secreted by the macrophages [127]. A somewhat different outcome of macrophage-to-fibroblast crosstalk was observed in another regeneration-friendly milieu, the heart of salamander. Cardiac cryoinjury in that case is followed by an efficient repair process leading to cardiomyocyte proliferation and regeneration. Depletion of macrophages inhibits repair by inducing alternative fibroblast activation that is characterized by distinct ECM production profile and increased collagen cross-linking by lysyl oxidase activity [128]. Regeneration impairment relies entirely on fibrosis, as cardiomyocyte proliferation remained unaffected, suggesting that the macrophage–fibroblast dipole is the main regulator of cardiac repair in this case.

In the mouse environment again, an IL18-mediated macrophage–fibroblast crosstalk was proposed to explain the pro-repair and antifibrotic action of curcumin in MI models (Figure 2). While treatment of neonatal cardiac fibroblasts with curcumin did not affect their fibrosis-related expression profile, their profibrotic response to TGFβ fibroblast exposure, although maintained when exposed to the secretome of LPS-stimulated RAW64.7 cells (a mouse macrophage-like cell line), was diminished when the macrophages were treated with curcumin [129]. Restoring IL18 expression in the macrophages sustained the profibrotic response.

The effect of macrophage status alteration has also been studied in various heterogeneous settings (Figure 3). As mentioned above, the M1/M2 stimulation scheme is not fully representative of the in vivo state; yet it is still valuable as a tool for understanding aspects of macrophage–fibroblast interactions in cardiac repair and fibrosis. Notably, a series of M2 stimulation subcategories (M2a, b, c, and d) have been reported to exert different actions concerning cardiac fibroblast activation and fibrosis. Ex vivo stimulation of mouse macrophages towards the M2b status appears to lead to a strong reparatory phenotype. Indeed, when the M2b-activated cells were directly injected into mouse infarcts, they stimulated repair by reducing PDGF-dependent activation of cardiac fibroblasts [130]. Interestingly, M2b (induced upon exposure to LPS combined with IgG) appears to be superior to other M2-polarization subtypes (M2a, M2c) in suppressing cardiac fibroblast proliferation, migration, collagen expression, and trans-differentiation to myofibroblasts, as suggested by direct comparison experiments in co-culture settings. On the other hand, inflammatory stimulation of macrophages may also be responsible for PDGF-dependent profibrotic activation of cardiac MSCs, giving rise to myofibroblasts that sustain fibrosis in experimental MI [131] (Figure 3). In a direct cardioprotective setting, BMDMs were stimulated to adopt the reparative M2b phenotype and back injected into rat hearts undergoing ischemia reperfusion. The antifibrotic and cardioprotective effects of the macrophage transfer were evident two weeks later and were associated with lower levels of members of the PDGF family (types C and D) as well as decreased levels of type α and β PDGF receptors and their phosphorylated forms by the fibroblasts [132]. The overall detainment of the otherwise activated PDGF pathway during ischemia/reperfusion (I/R) injury was recapitulated in vitro during co-culture of rat cardiac fibroblasts with M2b macrophages (or exposure to their secretomes), strongly suggesting that specifically this stimulation of macrophages may offer translational opportunities in ischemic cardiac fibrosis.

On the other hand, IL10 infusion in an MI model was shown to be beneficial for inducing fibroblast proliferation and migration to protect from post-MI ruptures [133]. IL10 acted by stimulating M2 polarization of macrophages, while mouse cardiac fibroblast cultures from IL10 receivers showed higher proliferation capacity and a lower collagen I/III expression ratio. Similarly, M2 polarization of macrophages was shown to occur post MI depending on the presence of V-set and immunoglobulin domain containing 4 (VSIG4), a complement receptor of the IgG superfamily that is being up-regulated in BMDMs by hypoxia inducible factor 1 subunit alpha (HIF1α) in ischemic conditions [134]. In this case, the M2 macrophages promote proliferation, migration, collagen expression, and myofibroblast trans-differentiation, acting beneficially by augmenting repair following MI (Figure 3). Conversely, VSIG deficiency leads to reduced M2 polarization and reduced production of IL10 and TGFβ in the MI hearts and in macrophage–fibroblast co-cultures, which is associated with impaired fibroblast pro-repair activities (compromised proliferation and, in that case, also collagen production and αSMA induction), leading to dysfunction and increased mortality due to MI rupture [134].

IL4, a major M2 polarization mediator, induced alternative macrophage polarization in MI but failed to do so in tribbles pseudokine-deficient mice (Trib^−/−^), an apparently important M2-regulator [135]. OPN and IL1α were suggested to be the mediators acting downstream of IL4 in that case, as they were secreted by IL4-polarized macrophages and induced myofibroblast trans-differentiation sustaining cardiac repair (Figure 3). In another approach, where BMDMs were M2 polarized by IL4 exposure, the extracellular vesicles (EVs) were isolated from the macrophage secretome and administered to mice, leading to increased fibrosis and augmented dysfunction early post MI (Figure 3). The deleterious action was attributed to CircUbe3a, a circular RNA contained in the EVs, which mediated the ability of the M2 polarized BMDMs to induce cardiac fibroblast proliferation and collagen overexpression in co-culture studies [136]. At variance, an IL4-exposed cardiac macrophage cell line suppressed proliferation and fibrotic expression of cardiac fibroblasts in a trans-well co-culture experiment [137]. In this setting, the fibroblasts were pre-exposed to hydrogen peroxide (H_2_O_2_), mimicking ischemic conditions in MI, and in contrast to the M2 (IL4)-activated macrophages, LPS-exposed proinflammatory macrophages exerted opposite effects, protecting the fibroblasts against apoptosis (Figure 3). Although direct cardiac injection of M1 macrophages in rat infarcts was shown to promote fibrosis (Figure 3) [137], the secreted mediators of these differential effects following distinct macrophage activation regimes have not been fully elucidated. In contrast, neuregulin overexpression by CD206^+^ macrophages isolated from murine hearts a week post MI seems to mediate their ability to protect cardiac fibroblasts against apoptosis and senescence [138]. The underlying mechanism involves binding of neuregulin to its receptors (Erb2 and Erb4) on the cell surface of fibroblasts and ERK/Akt signaling activation (Figure 3). Although CD206 expression is not exclusively confined to M2-macrophages in the cardiac tissue, this direct cardiac macrophage to fibroblast signaling setting could be interpreted as a M2-like activation-mediated profibrotic effect of the macrophages, being conformed with elevated post-MI fibrosis under conditions of their accumulation in cardiac tissue [138].

Another study directly examined the role of an E3 ubiquitin ligase, WWP2, in profibrotic macrophage stimulation [139]. Using the AngII infusion model to induce non-ischemic cardiac fibrosis, it was shown that WWP2 stabilized interferon regulatory factor 7 (IRF7) by non-degradable mono-ubiquitination, a process leading to induced expression of the chemokine CCL5. This allowed the prevalence of an M1-like phenotype by Ly6C^hi^ monocyte-derived macrophages infiltrating the infused hearts, which when co-cultured with cardiac fibroblasts, induced *Acta2* (encoding αSMA) expression and myofibroblast trans-differentiation of the latter (Figure 3). Furthermore, factors secreted from BMDMs with loss-of-function of WWP2 failed to induce profibrotic gene expression by the fibroblasts. This paracrine effect was confirmed by in vivo experimental settings were macrophage-specific WWP2 deletion resulted in reduced profibrotic gene expression and histological fibrosis (and dysfunction) in AngII-infused hearts [139]. Another member of the E3 family, tripartite motif containing protein 21 (TRIM21), also promotes M1 macrophage polarization, partly via the PI3K/Akt pathway. TRIM21 expression is up-regulated in macrophages post MI and mediates myocardial damage and apoptosis and adverse cardiac remodeling, while its overexpression in macrophage-like RAW264.7 cells inhibits cardiac fibroblast migration (Figure 3), likely explaining the increased post-MI repair observed in Trim21^−/−^ mice [140].

Overall, macrophage activation status differentially affects fibroblast responses during cardiac repair and fibrosis (Figure 3) and merits systematic study efforts to fully reveal the specific mediators involved that could form an arsenal for future intervention studies.

## 6. Cardiac Fibroblasts Affecting Macrophages

Following a cardiac injury, fibroblasts show high plasticity and may vividly alter their secretory profile, affecting other cell populations in a time-dependent manner [165]. Indeed, cardiac fibroblasts were shown to alter immune cell phenotypes, including macrophages, at multiple levels during injury and subsequent repair (Figure 4). The development of their own inflammatory phenotype is a crucial point in these processes. Thus, HF biopsy-derived fibroblasts from all cardiac chambers present with a strong proinflammatory cytokine production profile [166]. In mouse MI models, the fibroblasts secrete granulocyte-macrophage colony stimulating factor (GM-CSF), directing a myeloid-favorable differentiation process in bone marrow and leading to cardiac recruitment of monocytes and neutrophils that depend on CCL2 and CXCL2, respectively (Figure 4). The recruited myeloid cells compromise cardiac wall integrity by activating MMP-9 proteolysis and IL1β and IL6 inflammation [141]. A fraction of fibroblasts also secretes GM-CSF in autoimmune myocarditis settings and recruit proinflammatory Ly6C^hi^ monocytes in a process dependent on the presence of the proinflammatory IL17A cytokine [142]. IL17A loss-of-function, specifically in these Sca1+ fibroblasts, partially blocks HF development and mouse lethality also in MI. Similar fibroblasts appear to populate HF hearts, implying a more universal mechanism of cardiac degeneration by IL17A-mediated fibroblast pathways [143]. Moreover, in co-culture experiments, fibroblasts were shown to facilitate the differentiation of both proinflammatory Ly6C^hi^ and reparatory Ly6C^lo^ monocytes to macrophages. In a myocarditis milieu, fibroblasts respond to abundant IL17A and impair the ability of macrophages derived from Ly6C^hi^ monocytes to exert efferocytosis. While they also prevent reparatory Ly6C^lo^ monocytes from differentiating into reparatory MHCII+ macrophages, both events exacerbate inflammation and compromise the cardiac outcome [144] (Figure 4). In vitro settings recapitulated the proinflammatory recruitment of monocytes by activated cardiac fibroblasts. Indeed, the CCL2, CCL7, and CX3CL1 chemokines were found to be elevated in the secretomes of mouse cardiac fibroblasts upon TGFβ/IFNγ stimulation (Figure 4). Consequently, exposure of splenic monocytes in these secretomes led to superior recruitment of proinflammatory Ly6C^hi^ monocytes over reparative Ly6C^lo^ monocytes in a migration assay [145]. This altered recruitment potential of cardiac fibroblasts may affect remodeling features occurring in the so-called cardio–splenic axis that play an important role in the development of HF under chronic inflammation [146] and may even explain sex-dependent differences in fibrosis and repair. CCL2 and CCL7, to Cx3CL1 ratios are higher in cardiac fibroblast secretomes from male mice, conforming well with higher abundance of proinflammatory (CD206-negative) macrophages in the male hearts [147]. Sex-dependent differences in cardiac remodeling and repair are evident in human conditions and are recapitulated in animal models [167], and the fibroblast–macrophage dipole could also be critically affecting this aspect.

Profibrotic stimulation of fibroblasts also contributes to the development of a proinflammatory milieu in the cardiac tissue (Figure 4). This was shown artificially by fibroblast-specific transduction of profibrotic YAP expression that resulted in macrophage infiltration in murine hearts [148]. Similarly, specific loss of function of the Hippo pathway kinases Lats1 and Lats2 in fibroblasts unleashed their potential to trans-differentiate into myofibroblasts, canonically ascertained under intact Hippo signaling. This resulted in the formation of spontaneous fibrotic lesions in the absence of exogenous injuries that were accompanied by inflammatory macrophage (IFNIC) expansion, likely due to the up-regulation of proinflammatory cytokines in the mutated fibroblasts [149]. On the contrary, following in vitro AngII-induced trans-differentiation, cardiac myofibroblasts paracrinely induced apoptotic cell death of M1-polarized splenic macrophages and mediated the M2-like polarization of macrophage survivors [150]. A similar process appeared to occur after adoptive transfer of macrophages in a viral myocarditis setting that seems to depend on leptin production by the myofibroblasts, which may act as promoters of resolution of inflammation [150]. In another setting, AngII triggered a Toll/IL1 receptor domain containing adaptor inducing IFN-β (TRIF)-dependent cytokine overexpression by atrial fibroblasts, which in turn induced macrophage chemotaxis [151]. Conversely, murine macrophages further induced AngII-mediated atrial fibroblast proliferation, likely establishing a vicious circle promoting AngII-dependent atrial fibrillation and associated atrial fibrosis (Figure 4).

Moreover, antifibrotic pathways and effectors are activated in parallel to pro-fibrotic signaling during fibrotic response of cardiac fibroblasts, achieving detainment of fibrosis overextension partially through macrophage regulation (Figure 4). Indeed, myofibroblast-specific deletion of the inhibitory member of the SMAD/TGFβ1 axis, Smad7, in pressure overload resulted in increased cardiac fibrosis and dysfunction accompanied by MMP2-mediated collagen denaturation. The activated fibroblasts in this case paracrinely induced macrophage expansion in the murine hearts, which was collectively mediated by extracellular matrix protein 1 (ECM1), connective tissue growth factor (CTGF), secreted protein acidic and cysteine rich (SPARC), and other secreted factors, as revealed by proteomic analysis [152].

## 7. Macrophage to Fibroblast Conversion

Under some circumstances, macrophages may adopt a fibroblast-like phenotype. Macrophages can become collagen producers themselves, an identity canonically attributed to fibroblasts. This has been reported to occur in both zebrafish cryoinjured heart and murine MI conditions in a fibroblast-independent manner [168], while in the TAC model, macrophage-expressed SPARC may increase cardiac stiffness by processing pro-collagen into insoluble fibrillar collagen [169]. The ECM-associated protein, ECM-1, which has been shown to be deposited in fibrotic tissue in human HF, was found to be a product of both inflammatory and reparatory cardiac macrophages, as well as of fibroblasts in experimental MI [170]. ECM-1 stimulates in turn cardiac fibroblast collagen production and trans-differentiation into myofibroblasts, the latter by binding to LDL receptor related protein 1 (LRP1) cell surface receptor and Rho GTPase signaling activation [170]. In contrast, it inhibits fibroblast migration by lowering CCL2 expression [170]. Thus, macrophages may substitute fibroblasts in ECM production under cardiac injury conditions and affect critical fibroblast-mediated repair processes.

Interestingly, pathways that are normally attributed to fibroblast differential activation may also potentiate macrophage activities promoting a similar outcome. For example, YAP and TAZ, members of the Hippo pathway, whose activation is known to sustain profibrotic fibroblast activation, interact with histone deacetylase 3 (HDAC3) in BMDM to increase IL6 and decrease Arginase 1 expression, thus compromising their reparative phenotype. This also occurs in MI, conferring increased proinflammatory and decreased reparative potential to cardiac macrophages, and leading to higher profibrotic gene expression and stronger histological fibrosis in infarcted hearts [171]. Consequently, YAP/TAZ deficiency induces macrophage reparative phenotype formation and angiogenesis but reduces post-MI fibrosis conferring cardioprotection [171], while its myeloid-specific deletion or pharmacological inhibition retains adverse remodeling in pressure overload (TAC model) [172].

According to some reports, under pathological conditions, part of the activated fibroblasts mediating fibrosis may stem from the macrophages themselves. Indeed, when using scRNASeq in the AngII infusion model, a small cluster of macrophages was identified that expressed αSMA and transgelin, thus adopting a myofibroblast phenotype [173]. The presence of αSMA+ cells was further confirmed in lineage tracers expressed under *Ccr2*, *Cx3cr1*, and *Lyve1* promoters and, using parabiosis experiments, it was shown that this transition depends on the expression of the N6 methyladenosine demethylase ALKBH5 in circulating monocytes. Mechanistically, the up-regulated m6A RNA modifier ALKBH5 mediates in turn IL11 overexpression, thus promoting cardiac fibrosis and dysfunction during cardiac hypertension injury [173]. Such studies conform with previous ones showing that macrophages may adopt a fibroblast-like phenotype following prolonged culturing or in MI [174]. Notably, it appears that macrophage to fibroblast conversion may occur in parallel to “conventional” paracrine effects of macrophages on fibroblasts, as those described in the previous section. Indeed, in mouse ischemia-reperfusion injuries, a subset of inflammatory macrophages highly expressing S100a9 mediate NFkB and NLR family pyrin domain containing 3 (NLRP3) signaling in the early phase and TGFβ in the subsequent reparatory phase [72]. These S100a9-mediated processes render this macrophage subset capable of mediating both the fibroblast-to-myofibroblast differentiation by releasing TGFβ and the macrophage-to-myofibroblast transition by expressing αSMA on their own [72]. Both pathways restrict cardiac repair and lead to fibrosis.

It is worth noting, though, that the discrimination between macrophages and fibroblasts, not only in the heart but also in other organs, may be masked under some circumstances in mouse models due to the co-expression of some markers and other technical issues despite the use of fate mapping tools [175].

## 8. Third Party Involvement

More complex tripartite interactions between cardiac macrophages and fibroblasts include the involvement of cardiomyocytes or other cardiac cells and are crucial in cardiac repair regulation [176]. This is expected in an organ with multiple cell types actively involved in the injury-repair process. However, a detailed examination of the various bipartite interactions between fibroblasts or macrophages and other cell types in the context of cardiac repair is beyond the scope of the present review. Instead, a brief introduction including some characteristic third-party contributions is quoted here, while the reader can refer to several comprehensive reviews [177,178,179,180,181,182].

Both fibroblasts and macrophages communicate with cardiomyocytes affecting their function. The intercommunication between fibroblasts and cardiomyocytes can be mediated by connexin 43 (Cx43)-containing gap junctions, membrane nanotubes, or, in a paracrine manner, through secreted molecules [181]. Seemingly, macrophage secretion of amphiregulin controls Cx43 phosphorylation in cardiomyocytes and maintains gap junction homeostasis protecting against arrhythmias [183]. In the Dsg2^mut/mut^ model of arrhythmogenic cardiomyopathy, CCR2+ macrophages recruited via NFκB activation in cardiomyocytes affect both fibroblast and cardiomyocyte pathogenic profiles. Notably, the abundance of periostin+ profibrotic fibroblasts was dependent on CCR2+ macrophages and diminished in Ccr2^−/−^ mice [82], again suggesting a protective action of resident macrophages.

Like macrophages, other innate immune cells such as the polymorphonuclear neutrophils (PMNs) progressively adopt proinflammatory and reparatory phenotypes following an acute cardiac injury as MI [184]. In fact, efferocytotic ingestion of PMNs by macrophages induces the macrophage reparatory phenotype, while in parallel, PMNs instruct reparative polarization of macrophages in a paracrine manner by secreting neutrophil gelatinase-associated lipocalin (NGAL) [185]. The PMN phenotype and secretome become further differentiated in a time-dependent manner post MI and include the release of fibrinogen that may stimulate fibroblast proliferation at later time points [186]. Thus, PMN crosstalk with the macrophage–fibroblast dipole can affect cardiac repair [180]. Despite their reparative abilities, however, PMNs and detrimental PMN-derived NETosis (neutrophil extracellular trap formation) appear to promote CCR2+ macrophage infiltration and fibrosis in the long term post MI [187].

There are examples where cells of the adaptive immunity system also interfere in the macrophage–fibroblast partnership. In the characteristic case of pressure overload injury (TAC model), for instance, an initial increase in the levels of CCL2, CCL7, and CCL12 chemokines motivates both CCR2+ and CCR2- macrophages. The former, however, are responsible for T cell activation (both CD4+ and CD8+) and expansion in the heart and the lymph nodes [87]. This T cell and CCR2+ macrophage activation feature affects cardiomyocyte hypertrophy leading to systolic dysfunction, but it also stimulates fibrosis pathways increasing the cardiac expression of collagens and fibronectin.

## 9. Translational Opportunities

The macrophage–fibroblast dipole (or vice versa the fibroblast–macrophage dipole) is an interactive network regulating both homeostatic and disease conditions, including tissue repair and fibrosis, in many tissues and organs [188]. As such, it has been determined as a valuable therapeutic target to improve cardiac repair [189]. For the time being, however, most anti-fibrotic and pro-repair outcomes in clinical studies and practice arise from indirect effects of HF treatment, mostly targeting cardiomyocyte homeostasis [35]. Moreover, efforts separately targeting fibroblasts or macrophages have been extensively applied, generating multiple options for future therapeutic strategies. In vitro fibroblast-based library screens have been used, for instance, to reveal novel antifibrotic substances with beneficial action in cardiomyopathy settings [190]. In addition, several strategies have been proposed to manipulate fibroblasts’ profibrotic activity, including the induction of their dedifferentiation or apoptosis, as well as the forced reprogramming of the fibroblast identity to adopt characteristics and functions of other cell types [191]. Vice versa, beneficial fibroblast actions could be exploited to induce cardiac regeneration [192]. Current anti-fibrotic and pro-repair approaches in the heart consider the importance of the time point of the intervention, as well as the cross-organ efficiency of the anti-fibrotic agent being applied [193,194]. In parallel, immunomodulation including manipulation of macrophage functions has been the subject of intense research approaches, as it offers obvious opportunities to beneficially intervene in cardiac regeneration and repair, and in combating fibrosis [179,195]. Accordingly, several phase one to four clinical trials used agents that modulate, among other, macrophage functions and recruitment and were shown in several cases to ameliorate the cardiac outcome in the context of cardiovascular diseases, including myocardial repair (recently listed in [100]).

From the above, it is obvious that approaches targeting the macrophage–fibroblast dipole would open a wider window of opportunity. However, such efforts are still at the preclinical level of evaluation. Some intriguing examples are briefly introduced here.

In a seminal work, triggering inflammation with zymosan was shown to be equally efficient as a direct cardiac injection of bone marrow-derived mononuclear cells or cardiac progenitor cells in promoting repair and reducing fibrosis in an I/R injury mouse model [153]. This was attributed to the motivation of two macrophage populations (CCR2+ and CXC3R1+, respectively) in the murine heart. In ex vivo co-culture experiments, it appeared that the CCR2+ population induced the expression of collagen I and its cross-linker lysyl oxidase, as well as of αSMA by cardiac fibroblasts, while CXC3R1+ macrophages induced the expression of CTGF [153]. Both pathways converge in promoting cardiac healing at the early stages following the ischemic insult. However, a long-term stimulation of the cardiac fibroblasts would be expected to confer detrimental fibrotic events in this case.

One appealing approach to manipulate cardiac fibrosis could be the trans-differentiation of fibroblasts (reprogramming) into cardiomyocytes achieved ex vivo and in vivo with transduction protocols introducing the appropriate transcription factors into the fibroblast genome [196]. Among the obstacles for the efficient application of such an intervention is the limited ability of transplanted fibroblasts to follow the trans-differentiation program in an MI milieu. This limitation is probably due to a molecular interaction between the reprogramming agent GATA4 and the phosphorylated form of STAT1, the latter emerging due to interferon activation pathways in cardiac fibroblasts. IFNβ, secreted by macrophages, was a trigger for this response in ex vivo settings and macrophage depletion unleashed the reprogramming efficiency of fibroblasts in vivo [154]. Vice versa, the IFN-mediated activation of fibroblasts led to the secretion of CCL2, CCL7, and CCL12, which in turn mediated the recruitment of IFNβ-secreting macrophages, apparently establishing a vicious circle that would inhibit therapeutic reprogramming approaches.

From the above, it is obvious that the manipulation of macrophage (or fibroblast) secretome could also alter the functional response of fibroblasts (or macrophages) for the benefit of more efficient cardiac repair. The secretome potential (biomaterial) can be thus artificially modified towards this direction. In a relevant approach, macrophages cultured on degradable polar hydrophobic ionic polyurethane (D-PHI) preferentially secrete molecules promoting regeneration (agrin, annexin A5) rather than profibrotic ones (MMP7, TNFα) and mediate higher fibroblast migration but lower fibroblast-mediated collagen gel contraction in a paracrine manner [197], suggestive of an induced pro-repair potential.

Importantly, agents such as the sodium-glucose cotransporter-2 (SGLT2) inhibitors, whose administration in HF patients has been recently shown to be a very promising intervention strategy [198,199], show beneficial effects in both the fibroblast and the macrophage compartment in preclinical models [200,201], further underlining the importance of consideration of these two cell types in tandem when examining or planning modes to orchestrate cardiac repair.

## 10. Conclusions—Perspectives

Recent advances primarily based on innovative tools (single cell- or nucleus-based omics, fate mapping genetic tools in animal models, etc.) both at the preclinical level and human cardiac biopsy material analysis shed light into the biology, function, heterogeneity, and plasticity of fibroblasts and macrophages in the heart. Like many other organs, these cell populations proved to be comprised of subsets that change functionally during- and post-injury and differentially affect each other. The complex cellular interplay includes additional cell types, primarily the cardiomyocytes, while endothelial cells, mural cells, other innate immune cells (neutrophils and other granulocytes), as well as cells of the adaptive immunity intervene or respond to the fibroblast–macrophage dipole in the context of cardiac repair and fibrosis. Extracellular components, such as the ECM itself, may play additional roles [202,203]. Ex vivo systems that faithfully or closely model and recapitulate these events [204,205] and measure the outcome (organotypic cultures of cardiac slices [206,207,208], organoids, heart-on-a-chip, three-dimensional and induced pluripotent stem cell-based approaches [209,210,211,212]) are invaluable tools towards our understanding of these complex processes. The generation of experimental platforms that will globally address the effects and interactions of multiple cell types and mediators under specific injury conditions is an unmet need. Specific cardiac niches and structures (valves, coronary vasculature, atrioventricular node, epicardial tissue, and pericardial entities) need to be separately addressed while recognizing spatial modalities. Nevertheless, the strong contribution and interactions of macrophages and fibroblasts unequivocally place this dipole on top of the relevant research and approaches, eventually aiming to significantly improve the cardiac injury outcomes. In this context, the careful selection of the time points, mediators, and macrophage and/or fibroblast subtypes that will serve as targets for interventions is an absolute prerequisite for the successful application of therapeutic regimens.

## Figures and Tables

**Figure 1 biomolecules-14-01403-f001:**
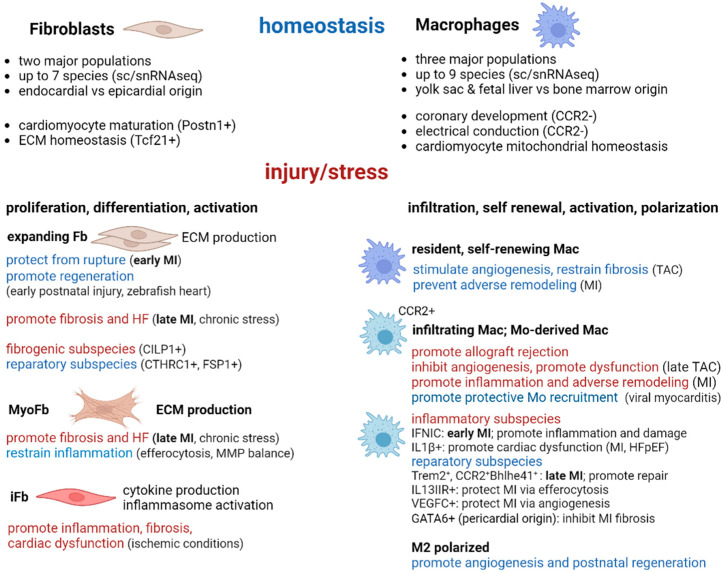
Main features of cardiac fibroblasts and macrophages. Major fibroblast and macrophage populations determine critical components of cardiac structure and function in homeostasis. Following injury, both compartments significantly change by proliferation, differentiation, activation, or recruitment of additional subsets from extracardiac resources. Depending on the condition and time point post injury (early vs. late), specific subpopulations may exert detrimental (red) or beneficial (blue) actions, further inhibiting or promoting, respectively, cardiac repair and regeneration. Fbs—cardiac fibroblasts; iFbs—inflammatory fibroblasts; MyoFbs—myofibroblasts; ECM, MMPs—extracellular matrix and metalloproteases, respectively; MI, TAC, HF, HFpEF—experimental models of myocardial infarction, thoracic aortic constriction, heart failure, and heart failure with preserved ejection fraction, respectively. sc/snRNASeq—single cell/single nucleus RNA sequencing. For the rest of the abbreviations and details, see text. Created with BioRender.com.

**Figure 2 biomolecules-14-01403-f002:**
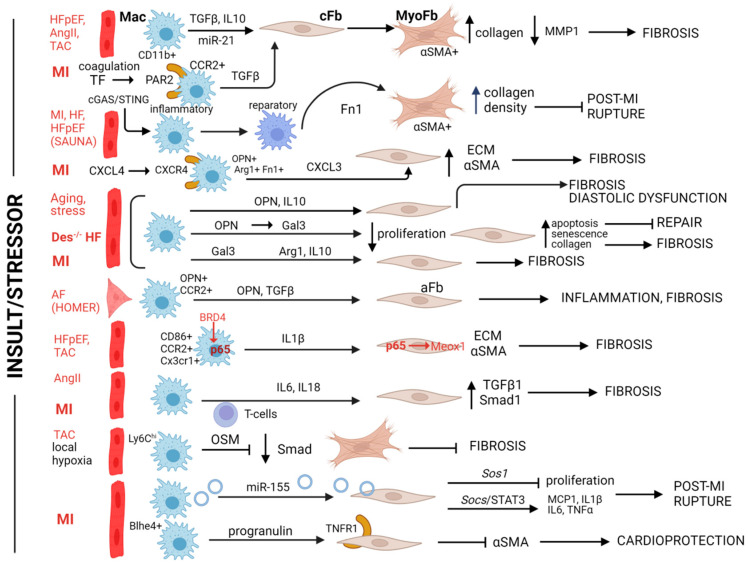
Macrophage-mediated effect on fibroblasts regulating cardiac repair and fibrosis. Depending on the insult/stress, primarily affecting ventricular or atrial cardiomyocytes (left side, in red), and pathways activated, macrophage subsets respond by altering their status and/or by secreting mediators that either promote (arrows with heads) or inhibit (arrows blunted) fibroblast activation, altering pathways regulating differentiation, ECM production, and cell survival. Depending on the pathology and the time point examined, this may support a beneficial (repair, inhibition of myocardial wall rupture) or a detrimental (fibrosis, promotion of myocardial wall rupture, cardiac dysfunction) outcome, crucial for heart failure (HF) development. Macs—macrophages; cFb—cardiac fibroblasts; aFbs—atrial fibroblasts; MyoFbs—myofibroblasts; MI, AngII, TAC, AF—myocardial infarction, AngII infusion, thoracic aortic constriction, and atrial fibrillation, respectively. HOMER, SAUNA, des^−/−^—animal models of cardiac injury and HF. For the rest of the abbreviations and details, see text. Created with BioRender.com.

**Figure 3 biomolecules-14-01403-f003:**
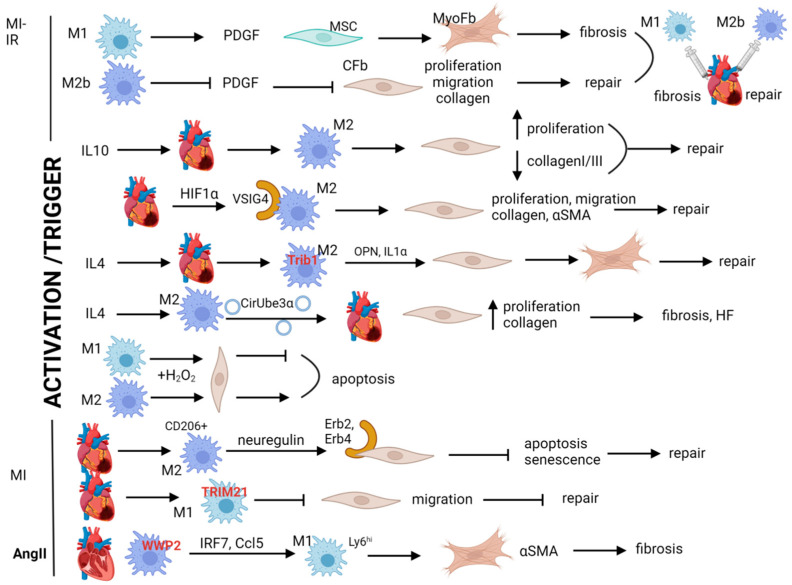
Macrophage status contributions to fibroblast-mediated cardiac repair and fibrosis. The in vivo and in vitro induction of the so-called M1 or M2 (and subcategories) macrophage activation affects the cardiac outcome in myocardial infarction (MI) or stress-induced (AngII) cardiomyopathy and HF models. In some cases, the pathways affected, and the mediators secreted by M1- or M2-activated macrophages that regulate in a paracrine manner cardiac fibroblast activation, differentiation, migration, proliferation, senescence, and survival, have been identified. Protocols using ex vivo activation and direct heart injection of macrophages have been applied to animal models (top right) leading to either beneficial repair or detrimental fibrosis. cFbs—cardiac fibroblasts; MyoFbs—myofibroblasts; MSCs—mesenchymal stem cells. Arrows with heads—induction; arrows blunted—inhibition. For the rest of the abbreviations and details, see text. Created with BioRender.com.

**Figure 4 biomolecules-14-01403-f004:**
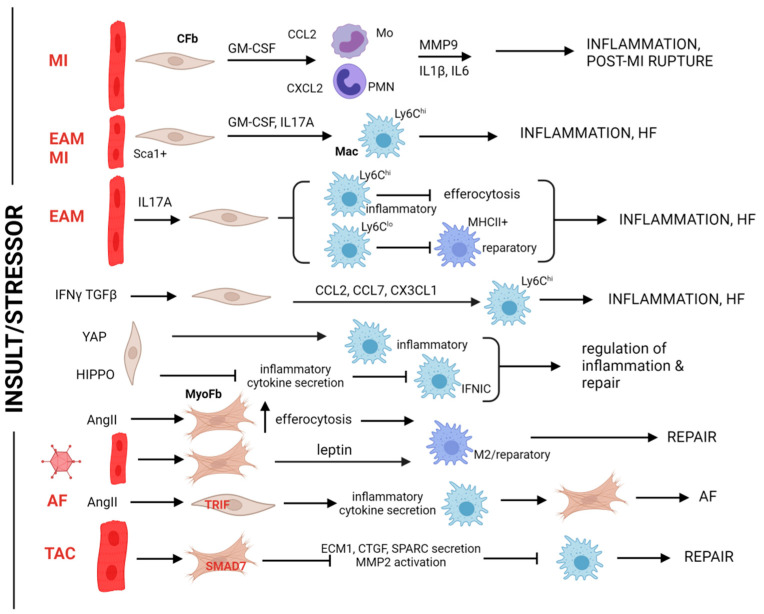
Fibroblast-mediated effects on macrophages regulating cardiac repair and fibrosis. Depending on the insult/stress, primarily affecting cardiomyocytes (left side, in red) and pathways activated, fibroblasts respond by altering their status and function, including differentiation into myofibroblasts, and/or by secreting mediators that either promote or inhibit the recruitment, activation, and differentiation of macrophages (and other innate immune cells). This, in turn, may affect the cardiac outcome promoting or inhibiting cardiac repair, fibrosis, and dysfunction depending on the pathology and the time point examined. Arrows with heads—induction; arrows blunted—inhibition; cFbs—cardiac fibroblasts; MyoFbs—myofibroblasts; Macs—macrophages; Mos—monocytes; PMNs—neutrophils; MI, EAM, AngII, Virus, AF, TAC—myocardial infarction, experimental autoimmune myocarditis, AngII infusion, viral myocarditis, atrial fibrillation, and thoracic aortic constriction, respectively. For the rest of the abbreviations and details, see text. Created with BioRender.com.

**Table 1 biomolecules-14-01403-t001:** Fibroblast populations and subsets in human and mouse hearts according to RNASeq.

Species	Main Protocol (Number of Cells/Nuclei)	Fb Subtypes	Fb Common Signature	Main Subpopulations and Signatures		Characteristics (Pathway Enrichment)	Ref.
Human	snRNASeq (363,213)	7 populations	*DCN*, *GSN*, *PDGFRA*	FB1	*SCN7A*, *BMPER*, *ACSM1*	Ventricular specificity. Canonical gene expression	[1]
				FB2	*CFH*, *ID4*, *KCNT2*	Atrial specificity. Canonical gene expression	
				FB3	*PTX3*, *OSMR*, *IL6ST*	Pronounced cytokine receptor expression	
				FB4	*POSTN*, *TNC*, *FAP*	Response to TGFβ	
				FB5	*FBLN2*, *PCOLCE2*, *LINC01133*	ECM regulation	
				FB6	*CD36*, *EGFLAM*, *FTL1*		
	snRNASeq (287,269)	3 clusters (FBI-III). 4 sub-clusters (FB-S1 to 4).	*DCN*, *ELN*	FB-S2	*NPPA*	Atrial cardiomyocyte marker	[2]
				FB-S3	*NOX4*, *IGF-1*	ECM regulation, fibrosis	
				FB-S4	*ADAMTS4*, *VCAN*, *AXL*	Profibrotic markers	
	scRNASeq (7495)	3 clusters	*DCN*, *C7*, *LUM*, *FBLN1*, *COL1A2*	FB1		ECM regulation, proliferation, atrial interactions	[8]
				FB2		Striated muscle tissue development	
	snRNASeq (704,296) ATAC-Seq (144,762)	6 clusters	*FBN1*, *SCARA5*, *ADAMTS2*, *COL6A3*, *PDGFRA*	FB4	*TNC*, COL1A2, COL3A1, COL8A1, FN1	Activated fibrotic sub-cluster	[4]
	snRNASeq	6 clusters		*CD10+*	*NEPRILYSIN*	Basal fibroblasts	[12]
				*Basal*		Basal fibroblasts	
				Activated	*POSTN*, COLIA2, COL3A1, MEOX1, AEBP1,	Fibrosis	
				MSC-like	*SHISA6*, LINC01133	Pluripotent pathway expression	
				Adventitial	*NR4A*, PPARG	Angiogenesis	
				Mechanical		Muscle contraction gene expression	
Human (MI)	snRNASeq (191,795); ATAC-RNASeq	4 clusters		Fib1	*SCARA5*, PCOLCE2		[7]
				Fib2	*PSOTN*, TNC, COL1A1, FN1, RUNX1	Fibrotic remodeling; myofibroblast identity	
				Fib3	*C7*, ABCA9		
				Fib4	*COL15A1*, SCN7A		
Human (DCM, HCM)	snRNASeq (592,689)	6 clusters		Activated Fbs	*POSTN*, NOX4, FAP, COL1A1, COL1A2, THBS4	Profibrotic	[6]
Human (HF)	snRNASeq (220,752) and scRNASeq (49,723)	9 populations	*DCN*, *LUM*, *CCDC80*, *FN1*, *NEGR1*, *ABCA8*, *CDH19* (*GPX3*, *PID1*, *TGFBR3*, *ACSM3*, *APOD: donor hearts*)	Fb1		Major basal subpopulation	[9]
				Fb2		Major basal subpopulation	
				Fb3	*GPX3*		
				Fb4	*PLA2G2A*		
				Fb5	*ELN*		
				Fb6	*TNC*		
				Fb7	*CCL2*		
				Fb8	*THBS4*, POSTN	Activated fibroblasts	
				Fb9	*SERPINE1*		
Mouse (MI)	scRNASeq (13,331 in healthy; 16,787 GFP^+^ cells in MI)	5 major clusters; 11 sub-clusters	*Col1a1*, *Pdgfra*	F-Act		Activated fibroblasts	[3]
				F-SH	*Ly6a (Sca1)* high		
				F-SL	*Ly6a* low		
				MYO	*Postn* and/or *Acta2*	Myofibroblasts	
				F-WNTX		Wnt signaling	
Mouse (MI)	scRNASeq (36,847)	3 populations (steady state)	*Wt1*	Type I	*Col1a1*, *Gsn*, *Dcn*		[10]
				Type II	*Wif1*, *Dkk3*		
				Type III	*Mt2*, *Timp1*		
				MyoFb	*Cthrc1*, *Acta2*, *Postn*	Myofibroblasts	

Major cardiac fibroblast clusters and subsets from the selected publications are listed. For the full lists and signatures, the reader should refer to the references provided. RNASeq—RNA sequencing; sc—single cell; sn—single nucleus; ATAC—Assay for Transposase-Accessible Chromatin; MI, DCM, HCM, HF—Myocardial infarction, dilated cardiomyopathy, hypertrophic cardiomyopathy, heart failure, respectively.

**Table 2 biomolecules-14-01403-t002:** Macrophage–fibroblast interactions that affect cardiac repair and fibrosis.

Condition	Effector Cell	Target Cell	Mediators-Pathways Involved	Outcome	Ref.
MI; *Ndufs4*^−/−^	Mac	cFb	Mitochondrial dysfunction	Impaired Fb proliferation, MyoFb formation, and repair	[108]
TAC; *Nlrc5*^−/−^	Mac	cFb	Hsp8, IL6	Increased MyoFb formation and fibrosis	[111]
AngII, HFpEF	CD11b^+^ Mac	cFb	TGFβ, MMP-1	Increased collagen production, MyoFb formation, and fibrosis	[112,113]
MI, HF	CCR2^+^ Mac	cFb	Coagulation, TGFβ	Fibrosis	[114]
TAC	Mac	cFb	TGFβ, IL10	Fibrosis	[88]
MI (early)	iMac	cFb	cGAS/STING, Fn1	MyoFb formation and repair	[70]
MI	Spp1^+^ Fn1^+^ Arg1^+^ Mac	cFb	CXCL4	Fb activation and fibrosis	[115,116]
HFpEF (SAUNA)	Mac	cFb	CXCL3	MyoFb formation and fibrosis	[117]
Stress; Aging	Mac	cFb	OPN, IL-10	Fibrosis; diastolic dysfunction	[118]
des^−/−^ HF	Mac	cFb	OPN, Gal3	Reduced repair activities; Fibrosis; HF	[22,119]
AF (HOMER)	Spp1^+^ CCR2^+^ Mac	aFb	TGFβ	Atrial fibrosis and fibrillation	[120]
HFpEF (HFD), TAC	CD86^+^ CCR2^+^ Cx3Cr1^+^ Mac	iFb	IL1β, BRD4, MEOX1, TGFβ	MyoFb formation and fibrosis	[121,122]
AngII	Mac	cFb	IL6	Fibrosis	[123]
TAC	Mac	cFb	miR-21	MyoFb formation and fibrosis	[124]
MI	Mac	cFb	miR-155, Sos1,	Inhibition of Fb proliferation; cardiac rupture	[125]
MI	Mac	cFb	miR-155, Socs1	Inflammation; cardiac rupture	[125]
TAC (early)	LY6C^hi^ Mac	cFb	OSM	Inhibition of MyoFb formation; reduced fibrosis	[126]
MI	Bhlhe4^+^ Mac	cFb	Progranulin, TNFR1	Inhibition of MyoFb formation; repair	[69]
Pharmacological injury (zebrafish larvae)	tnfa-neg Mac	cFb	adra1, midkine, collagen 12	Regeneration	[127]
cryoinjury (salamander)	Mac	cFb	Lysyl oxidase	Moderate collagen cross linking; repair/regeneration	[128]
MI	Mac	cFb	IL18	Fibrosis	[129]
MI; I/R	M2b Mac	cFb, MSCs	PDGF	Reduced PDGF signaling; repair	[130,131,132]
MI (with IL-10 infusion)	M2 Mac	cFb	IL10	Fb proliferation; reduced collagen I/III ratio	[133]
MI	M2 Mac	cFb	HIF1α, VSIG4, IL10, TGFβ	Fb proliferation, collagen expression, MyoFb formation; repair	[134]
ΜΙ	M2 Mac	cFb	Trib; IL4, OPN, IL10	MyoFb formation; repair	[135]
MI, IL4	M2 Mac	cFb	CircUbe3a	Fb proliferation, collagen expression; fibrosis; HF	[136]
Ischemia (H_2_O_2_)	M2 Mac	cFb	-	Fb apoptosis	[137]
Ischemia (H_2_O_2_)	M1 Mac	cFb	-	Fb protection	[137]
MI	CD206^+^ Mac	cFb	Neuregulin, ERK/Akt	Fb protection from apoptosis and senescence; fibrosis	[138]
AngII	M1 Ly6^hi^ Mac	cFb	WWP2, CCL5	MyoFb formation; fibrosis	[139]
MI	Mac	cFb	TRIM21, Pi3K/Akt	Inhibition of Fb migration; adverse remodeling	[140]
MI	cFb	Mo	GM-CSF, IL6, IL1β, MMP-9	Mo and PMN recruitment; cardiac rupture	[141]
ΕAΜ, MI	Sca1^+^ cFb	Mo	IL17A	Ly6^hi^ Mo recruitment; cardiac rupture; HF	[142,143]
EAM	cFb	Mo	IL17A,	Inhibition of efferocytosis; inhibition of reparatory Mac formation; inflammation	[144]
TGFβ/IFNγ stimulation	cFb	Mo	CCL2, CCL7, CX3CL1	Ly6^hi^ Mo recruitment; inflammation	[145,146,147]
MI	cFb	Mac	YAP	Inflammatory Mac infiltration	[148]
MI	MyoFb	Mac	Hippo	Regulation of IFNIC Mac infiltration; repair	[149]
AngII, viral myocarditis	MyoFb	Mac	leptin	Mac M2 polarization; resolution of inflammation	[150]
AngII	aFb	Mac	TRIF	Atrial fibrosis and fibrillation	[151]
TAC	MyoFb	Mac	Smad7, MMP2	Detainment of Mac expansion, cardiac fibrosis, and HF	[152]
I/R, zymosan	CCR2^+^ Mac	cFb	Lysyl oxidase	Formation of MyoFb; repair	[153]
I/R, zymosan	CXC3R1^+^ Mac	cFb	CTGF	Activation of Fb; repair	[153]
MI	Mac	Fb	IFNβ, GATA4, STAT1	Inhibition of Fb to cardiomyocyte trans-differentiation; repair failure	[154]

Macs, Mos—macrophages, monocytes, respectively; M1, M2, M2b—macrophage activation status; cFb, aFb, iFb, MyoFb, MSC—cardiac fibroblasts, atrial fibroblasts, induced fibroblasts, myofibroblasts, mesenchymal stem cells, respectively; MI, I/R, AngII, TAC, HFD, EAM—myocardial infarction, ischemia reperfusion, AngII infusion, thoracic aortic constriction, high-fat diet, and experimental autoimmune myocarditis, respectively; cardiac injury models. For the rest of the abbreviations, see text.

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
