# Peer review of "The Macrophage–Fibroblast Dipole in the Context of Cardiac Repair and Fibrosis"

_biomolecules, 2024, doi:10.3390/biom14111403_

Round 1

Reviewer 1 Report

Comments and Suggestions for Authors

The review article was extremely well-written and well organized. There are very few suggestions that would improve the depth or breadth of the manuscript. 

Author Response

Revewer1

The review article was extremely well-written and well organized. There are very few suggestions that would improve the depth or breadth of the manuscript. 

Thank you for your positive comments. To improve depth and breadth a new figure (Figure 1) and a new Table (Table 1) were included to enforce the reader’s information while some aspects of macrophage biology regarding cardiac repair were further elaborated. Moreover, additions were made to expand the description of macrophage roles in cardiac repair and fibrosis, and translational opportunities were further discussed.

Reviewer 2 Report

Comments and Suggestions for Authors

This review by Psarras S et al., is extensive and resourceful review on the role of macrophage and fibroblast in cardiac tissue damage and regeneration. Each section in this review is easy to follow, this review is well written. I have only minor suggestion: the tables are information, however, I would suggest to add 2-3 graphical representation of cardiac repair (eg. MI model) in which authors could represent fibroblast and macrophages. These figures will attractive as well informative for the readers.

This review can be accepted for publication.

Comments on the Quality of English Language

Maybe minor english editing is required

Author Response

Reviewer 2

This review by Psarras S et al., is extensive and resourceful review on the role of macrophage and fibroblast in cardiac tissue damage and regeneration. Each section in this review is easy to follow, this review is well written. I have only minor suggestion: the tables are information, however, I would suggest to add 2-3 graphical representation of cardiac repair (eg. MI model) in which authors could represent fibroblast and macrophages. These figures will attractive as well informative for the readers.

This review can be accepted for publication.

Maybe minor english editing is required

Answer:

Thank you for your constructive comments. A new figure has now been introduced (Figure 1) to summarize major features of each cell type in the context of cardiac fibrosis and repair. This schematic includes features corresponding to MI as well as to other major types of cardiac injury (chronic stress, myocarditis) to globally represent the contribution of both cell types. In addition, a list of major fibroblast subtypes suggested by recent RNA sequencing experiments conducted at the single cell level is included (Table 1). Finally, careful English editing has been thoroughly applied to improve the text.

Reviewer 3 Report

Comments and Suggestions for Authors

Dear authors,

Thank you for submitting the manuscript “The macrophage-fibroblast dipole in the context of cardiac repair and fibrosis”.

The main question addressed by the research was to cover the multifaceted aspects and crosstalk of macrophage in heart disease interventions. The topic is original because the search for new targets involved in the pathogenesis of repair and fibrosis may be crucial in clinical implementation. The results are mostly presented clearly and accurately. The figures and tables correspond to the description in the text, are well designed and reflect important information. I think that this is a very worthy work. The references are appropriate. I express my gratitude to the authors for their work and my great pleasure in reading their results. The article may be published in its current version.

Author Response

Reviewer 3

Thank you for submitting the manuscript “The macrophage-fibroblast dipole in the context of cardiac repair and fibrosis”.

The main question addressed by the research was to cover the multifaceted aspects and crosstalk of macrophage in heart disease interventions. The topic is original because the search for new targets involved in the pathogenesis of repair and fibrosis may be crucial in clinical implementation. The results are mostly presented clearly and accurately. The figures and tables correspond to the description in the text, are well designed and reflect important information. I think that this is a very worthy work. The references are appropriate. I express my gratitude to the authors for their work and my great pleasure in reading their results. The article may be published in its current version.

Answer:

Thank you for your positive comments.

Reviewer 4 Report

Comments and Suggestions for Authors

Many review articles have already discussed the role of macrophages in myocardial regeneration and fibrosis.To stand out by providing a unique perspective and focusing on cutting-edge research trends is important. Authors could consider writing your review from the following innovative perspectives:

1. Temporal dimension: Elaborate on the functional differences of macrophages at different stages of myocardial regeneration and fibrosis, especially focusing on their roles in early injury response versus late tissue repair.

2. Heterogeneity of macrophages: Delve into the different subtypes of macrophages (such as M1 and M2 macrophages) and their distinct roles in myocardial regeneration and fibrosis after injury, highlighting the current understanding of the dynamic changes in macrophage subpopulations.

3. Cellular communication and signaling pathways: Utilize the latest single-cell sequencing technology to explain the interactions between macrophages and other cells (such as cardiomyocytes, fibroblasts, and endothelial cells), especially how specific signaling pathways regulate myocardial regeneration and fibrosis.

4. Metabolic state of macrophages: Explore the metabolic changes in macrophages during myocardial injury and repair, especially how metabolic reprogramming affects their functions, influencing myocardial regeneration and fibrosis.

5. Therapeutic targets and clinical potential: Analyze how modulating macrophage functions could be used to intervene in myocardial regeneration and fibrosis, drawing on recent advances in drug development or gene therapy to present potential clinical applications.

6. Systemic diseases and personalized treatment: Discuss whether macrophage behavior in myocardial regeneration and fibrosis differs in patients with systemic diseases such as diabetes or hypertension, and explore the possibilities of personalized treatment.

Comments on the Quality of English Language

The quality of the writing in English needs improvement.

Author Response

Reviewer 4

Many review articles have already discussed the role of macrophages in myocardial regeneration and fibrosis. To stand out by providing a unique perspective and focusing on cutting-edge research trends is important. Authors could consider writing your review from the following innovative perspectives:

Answer:

Thank you for your comments. This review examines the roles of both macrophages and fibroblasts in cardiac repair and fibrosis, mainly aiming at summarizing the effect of their strong and reciprocal interaction within the frame of these processes. Accordingly, to describe this unique view, the term dipole has been introduced in the review’s title. As such, the descriptions of each cell type’s contribution should not be too extensive, and they should be considered instead as minimally required introductions to support a better understanding of the dipole’s role and effects. With that in mind, detailed presentations of cardiac macrophage biology were largely omitted in favor of a more detailed presentation of their functions that are specifically interrelated with cardiac fibroblasts, and vice versa.

However, the comments are well appreciated, and additions of text, literature, as well as of a new figure (Figure 1) were made to conform with a broader macrophage presentation within this context. The additions or extensions are reported in detail underneath in a point-by-point answer format. Notably, some of the suggestions were already addressed to some extent in the original manuscript, and when applicable these cases are also mentioned in the point-by-point answers. Please note that the corresponding lines have changed in the revised version. New additions are marked with yellow in the revised version.

  1. Temporal dimension: Elaborate on the functional differences of macrophages at different stages of myocardial regeneration and fibrosis, especially focusing on their roles in early injury response versus late tissue repair.

Answer:

Timely alterations in macrophage functions in the context of cardiac repair (including regeneration) and fibrosis were already mentioned (lines 241-258 and 272-283 of the original version).

In the revised version, an addition (lines 377-412) along with 8 new references (both in yellow) supports a broader macrophage presentation focusing on altered macrophage features and functions during early and late post injury time points along different conditions. Furthermore, it is stated that The precise identification and characterization of the macrophage subsets at different time points post injury and in human cardiac disease is thus of particular importance as it should provide specific targets for future therapeutic interventions. Its detailed description is however beyond the scope of this review and the reader should refer to several comprehensive reviews summarizing current efforts”. (lines 399-407). Moreover, a figure has been added (Figure 1) to describe main macrophage features important for cardiac repair and fibrosis in a time-dependent manner (early vs late). Following the overall concept of this review these features are presented along with relevant fibroblast-specific ones in a comparative manner.

  1. Heterogeneity of macrophages: Delve into the different subtypes of macrophages (such as M1 and M2 macrophages) and their distinct roles in myocardial regeneration and fibrosis after injury, highlighting the current understanding of the dynamic changes in macrophage subpopulations.

Answer:

The heterogeneity of macrophages present or emerging in the cardiac tissue has been described in lines 228-267 of the original version (including information from publications using scRNASeq or/and snRNASeq data). The macrophage status alterations and their contribution in cardiac regeneration or fibrosis have been addressed, while the effects of their M1/M2 related plasticity were extensively handled (lines 371-402 and 649-729 of the original version, respectively).

In the new text added (lines 377-412 of the revised version), already described above, as well as in the new figure (Figure 1) the heterogeneity of macrophages and the distinct roles of activated forms are further elaborated, in addition to the previous descriptions.   

  1. Cellular communication and signaling pathways: Utilize the latest single-cell sequencing technology to explain the interactions between macrophages and other cells (such as cardiomyocytes, fibroblasts, and endothelial cells), especially how specific signaling pathways regulate myocardial regeneration and fibrosis.

Answer:

As mentioned in section 8 of the original version (“Third party involvement”, lines 944-951) the interaction of macrophages (or fibroblasts) with other cardiac cells such as cardiomyocytes and endothelial cells is beyond the scope of this review and the interested reader is encouraged to refer to several comprehensive reports (Refs 166-171). The interactions of macrophages with fibroblasts in the context of cardiac repair and fibrosis as the scope of the review per se have been addressed at several points throughout the text and scRNASeq-based suggestions were mentioned (e.g. lines 466-483, 565-572 and 861-872 of the original version).

  1. Metabolic state of macrophages: Explore the metabolic changes in macrophages during myocardial injury and repair, especially how metabolic reprogramming affects their functions, influencing myocardial regeneration and fibrosis.

Answer:

The role of the metabolic remodeling of macrophages including timely changes in cardiac repair has been addressed (lines 403-424 of the original version). In the revised version, the importance of metabolic remodeling in macrophage plasticity and contribution in cardiac repair and fibrosis has been further elaborated via an added text (lines 466-479) and additional references.

  1. Therapeutic targets and clinical potential: Analyze how modulating macrophage functions could be used to intervene in myocardial regeneration and fibrosis, drawing on recent advances in drug development or gene therapy to present potential clinical applications.

Answer:

Relevant translational opportunities were already mentioned in a separate section (section 9) by describing important intervention strategies affecting both macrophages and fibroblasts. Moreover, additional comments and suggestions were included under “Conclusions and Perspectives”. A detailed description of such approaches specifically focusing on macrophages is beyond the scope of the review. To further elaborate on relevant intervention strategies in regeneration and fibrosis a text has been added in section 9, referring to current fibroblast- and macrophage-specific modulation (lines 994-1014). In addition, perspectives of the benefits offered by the consideration of these two cell types in tandem are described in this very section (lines 1046-1051).

  1. Systemic diseases and personalized treatment: Discuss whether macrophage behavior in myocardial regeneration and fibrosis differs in patients with systemic diseases such as diabetes or hypertension, and explore the possibilities of personalized treatment.

Answer:

The elaboration of the macrophage behavior and effect at the clinical level and specifically in the context of systemic diseases (diabetes, hypertension, etc) is far beyond the scope of this article, especially regarding the exploration of the possibilities for personalized treatment. The review aims instead to reflect the strong interrelations of these two cellular entities as it is deduced from current research and to refer to relevant intervention approaches in cardiac repair and fibrosis.

Thank you again for your comments.

Finally, careful English editing has been thoroughly applied to improve the text.

Round 2

Reviewer 4 Report

Comments and Suggestions for Authors

Review comments:

The authors comprehensively summarized the role of stromal and immune cells in cardiac repair, as these cells significantly impact the heart’s recovery after injury or stress. Unfortunately, the heart’s complex repair processes often fail, leading to irreversible conditions like heart failure. Fibrosis, a scarring response involving collagen overproduction by cardiac fibroblasts, contributes to heart dysfunction. Macrophages, another immune cell type, can either support or hinder repair, depending on their state. The study highlights how understanding these cell interactions might open new therapeutic strategies for heart disease.

The authors have appropriately addressed the comments raised during the last review process, and the manuscript is now suitable for acceptance.